META-RESEARCH ARTICLE

# The evolving role of preprints in the dissemination of COVID-19 research and their impact on the science communication landscape

Nicholas Fraser[1]ʘ, Liam Brierley[2]ʘ, Gautam Dey[3,4], Jessica K. Polka[5], Máté Pálfy[6], Federico Nanni[7], Jonathon Alexis Coates[8,9]*

**1** Leibniz Information Centre for Economics, Kiel, Germany, **2** Department of Health Data Science, University of Liverpool, Liverpool, United Kingdom, **3** MRC Lab for Molecular Cell Biology, UCL, London, United Kingdom, **4** Cell Biology and Biophysics Unit, European Molecular Biology Laboratory, Heidelberg, Germany, **5** ASAPbio, San Francisco, California, United States of America, **6** The Company of Biologists, Cambridge, United Kingdom, **7** The Alan Turing Institute, London, United Kingdom, **8** Hughes Hall College, University of Cambridge, Cambridge, United Kingdom, **9** William Harvey Research Institute, Charterhouse Square Barts and the London School of Medicine and Dentistry Queen Mary University of London, London, United Kingdom

ʘ These authors contributed equally to this work.
* jonathon.coates@qmul.ac.uk

**Data Availability Statement:** All data and code used in this study are available on GitHub (https://github.com/preprinting-a-pandemic/pandemic_preprints) and Zenodo (DOI: 10.5281/zenodo.4501924).

## Abstract

The world continues to face a life-threatening viral pandemic. The virus underlying the Coronavirus Disease 2019 (COVID-19), Severe Acute Respiratory Syndrome Coronavirus 2 (SARS-CoV-2), has caused over 98 million confirmed cases and 2.2 million deaths since January 2020. Although the most recent respiratory viral pandemic swept the globe only a decade ago, the way science operates and responds to current events has experienced a cultural shift in the interim. The scientific community has responded rapidly to the COVID-19 pandemic, releasing over 125,000 COVID-19–related scientific articles within 10 months of the first confirmed case, of which more than 30,000 were hosted by preprint servers. We focused our analysis on bioRxiv and medRxiv, 2 growing preprint servers for biomedical research, investigating the attributes of COVID-19 preprints, their access and usage rates, as well as characteristics of their propagation on online platforms. Our data provide evidence for increased scientific and public engagement with preprints related to COVID-19 (COVID-19 preprints are accessed more, cited more, and shared more on various online platforms than non-COVID-19 preprints), as well as changes in the use of preprints by journalists and policymakers. We also find evidence for changes in preprinting and publishing behaviour: COVID-19 preprints are shorter and reviewed faster. Our results highlight the unprecedented role of preprints and preprint servers in the dissemination of COVID-19 science and the impact of the pandemic on the scientific communication landscape.

## Introduction

Since January 2020, the world has been gripped by the Coronavirus Disease 2019 (COVID-19) outbreak, which has escalated to pandemic status, and caused over 98 million cases and 2.1

**Funding:** NF acknowledges funding from the German Federal Ministry for Education and Research, grant numbers 01PU17005B (OASE) and 01PU17011D (QuaMedFo). LB acknowledges funding from a Medical Research Council Skills Development Fellowship award, grant number MR/T027355/1. The funders had no role in study design, data collection and analysis, decision to publish, or preparation of the manuscript.

**Competing interests:** I have read the journal's policy and the authors of this manuscript have the following competing interests: JP is the executive director of ASAPbio, a non-profit organization promoting the productive use of preprints in the life sciences. GD is a bioRxiv Affiliate, part of a volunteer group of scientists that screen preprints deposited on the bioRxiv server. MP is the community manager for preLights, a non-profit preprint highlighting service. GD and JAC are contributors to preLights and ASAPBio fellows.

**Abbreviations:** AAAS, American Association for the Advancement of Science; ACE2, angiotensin converting enzyme 2; API, Application Programming Interface; COVID-19, Coronavirus Disease 2019; CSHL, Cold Spring Harbor Laboratory; ECDC, European Centre for Disease Prevention and Control; HSD, honest significant difference; MERS, Middle East Respiratory Syndrome; ROR, Research Organisation Registry; SARS-CoV-2, Severe Acute Respiratory Syndrome Coronavirus 2; UK POST, United Kingdom Parliamentary Office of Science and Technology; WHO SB, World Health Organization Scientific Briefs.

million deaths (43 million cases and 1.1 million deaths within 10 months of the first reported case) [1–3]. The causative pathogen was rapidly identified as a novel virus within the family Coronaviridae and was named Severe Acute Respiratory Syndrome Coronavirus 2 (SARS-CoV-2) [4]. Although multiple coronaviruses are ubiquitous among humans and cause only mild disease, epidemics of newly emerging coronaviruses were previously observed in SARS in 2002 [5] and Middle East Respiratory Syndrome (MERS) in 2012 [6]. The unprecedented extent and rate of spread of COVID-19 has created a critical global health emergency, and academic communities have raced to respond through research developments.

New scholarly research has traditionally been communicated via published journal articles or conference presentations. The traditional journal publishing process involves the submission of manuscripts by authors to an individual journal, which then organises peer review, the process in which other scientists ("peers") are invited to scrutinise the manuscript and determine its suitability for publication. Authors often conduct additional experiments or analyses to address the reviewers' concerns in 1 or more revisions. Even after this lengthy process is concluded, almost half of submissions are rejected and require resubmission to a different journal [7]. The entire publishing timeline from submission to acceptance is estimated to take approximately 6 months in the life sciences [8,9]; the median time between the date a preprint is posted and the date on which the first DOI of a journal article is registered is 166 days in the life sciences [8].

Preprints are publicly accessible scholarly manuscripts that have not yet been certified by peer review and have been used in some disciplines, such as physics, for communicating scientific results for over 30 years [10]. In 2013, 2 new preprint initiatives for the biological sciences launched: PeerJ Preprints, from the publisher PeerJ, and bioRxiv, from Cold Spring Harbor Laboratory (CSHL). The latter established partnerships with journals that enabled simultaneous preprint posting at the time of submission [11]. More recently, CSHL, in collaboration with Yale and BMJ, launched medRxiv, a preprint server for the medical sciences [12]. Preprint platforms serving the life sciences have subsequently flourished, and preprints submissions continue to grow year on year; two-thirds of these preprints are eventually published in peer-reviewed journals [8].

While funders and institutions explicitly encouraged prepublication data sharing in the context of the recent Zika and Ebola virus disease outbreaks [13], usage of preprints remained modest through these epidemics [14]. The COVID-19 crisis represents the first time that preprints have been widely used outside of specific communities to communicate during an epidemic.

We assessed the role of preprints in the communication of COVID-19 research in the first 10 months of the pandemic, between January 1 and October 31, 2020. We found that preprint servers hosted almost 25% of COVID-19–related science, that these COVID-19 preprints were being accessed and downloaded in far greater volume than other preprints on the same servers, and that these were widely shared across multiple online platforms. Moreover, we determined that COVID-19 preprints are shorter and are published in journals with a shorter delay following posting than their non-COVID-19 counterparts. Taken together, our data demonstrate the importance of rapidly and openly sharing science in the context of a global pandemic and the essential role of preprints in this endeavour.

## Results

### COVID-19 preprints were posted early in the pandemic and represent a significant proportion of the COVID-19 literature

The COVID-19 pandemic has rapidly spread across the globe, from 3 patients in the city of Wuhan on the December 27, 2019 to over 46.1 million confirmed cases worldwide by the end of October 2020 (Fig 1A). The scientific community responded rapidly as soon as COVID-19

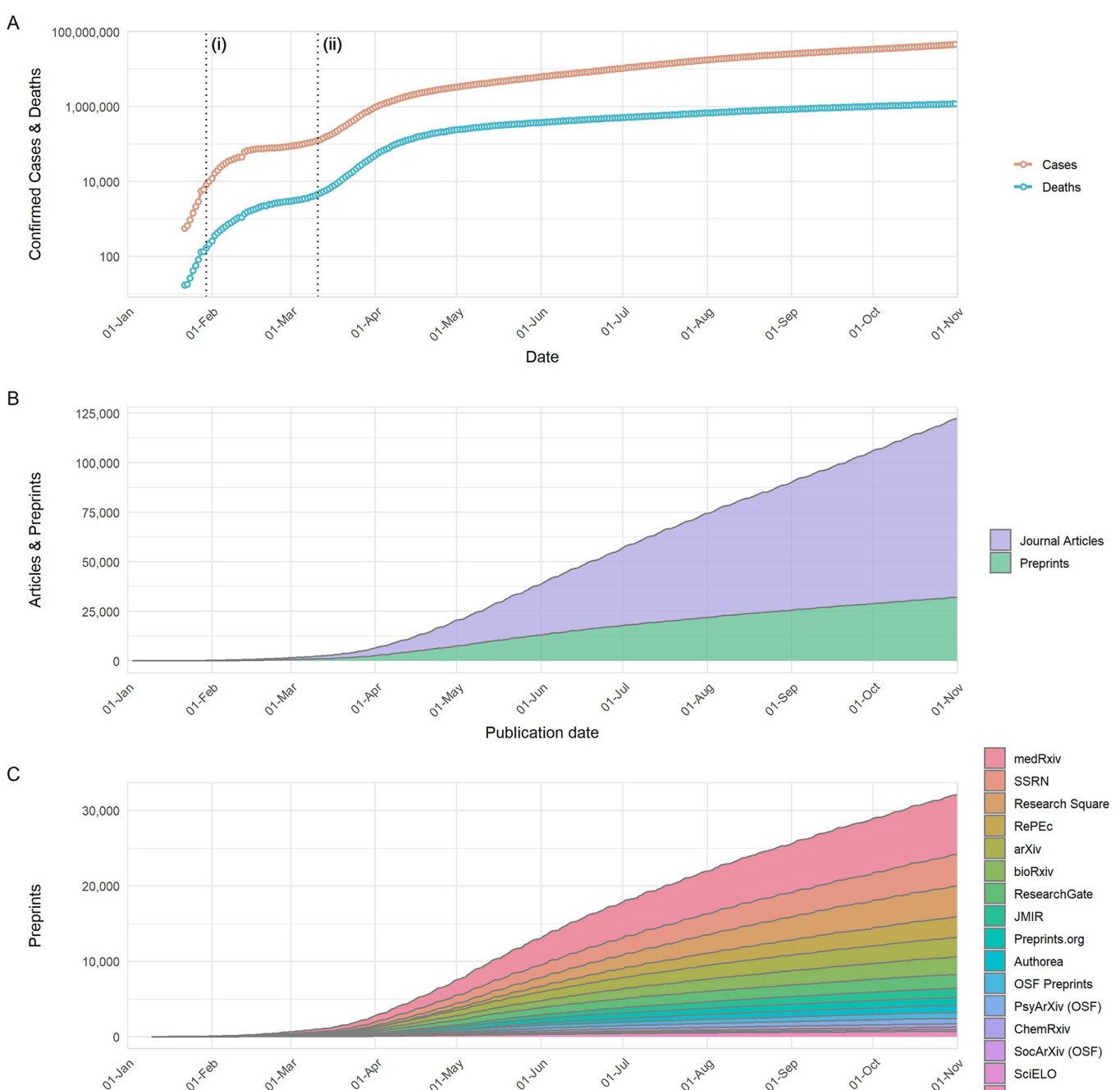

**Fig 1. Development of COVID-19 and publication response from January 1 to October 31, 2020.** (A) Number of COVID-19 confirmed cases and reported deaths. Data are sourced from https://github.com/datasets/covid-19/, based on case and death data aggregated by the Johns Hopkins University Center for Systems Science and Engineering (https://systems.jhu.edu/). Vertical lines labelled (i) and (ii) refer to the date on which the WHO declared COVID-19 outbreak a Public Health Emergency of International Concern, and the date on which the WHO declared the COVID-19 outbreak to be a pandemic, respectively. (B) Cumulative growth of journal articles and preprints containing COVID-19–related search terms. (C) Cumulative growth of preprints containing COVID-19–related search terms, categorised by individual preprint servers. Journal article data in (B) are based upon data extracted from Dimensions (https://www.dimensions.ai; see Methods section for further details), and preprint data in (B) and (C) are based upon data gathered by Fraser and Kramer (2020). The data underlying this figure may be found in https://github.com/preprinting-a-pandemic/pandemic_preprints and https://zenodo.org/record/4587214#.YEN22Hmnx9A. COVID-19, Coronavirus Disease 2019; WHO, World Health Organization.

emerged as a serious threat, with publications appearing within weeks of the first reported cases (Fig 1B). By the end of April 2020, over 19,000 scientific publications had appeared, published both in scientific journals (12,679; approximately 65%) and on preprint servers (6,710; approximately 35%) (Fig 1B)—in some cases, preprints had already been published in journals during this time period and thus contribute to the counts of both sources. Over the following months, the total number of COVID-19–related publications increased approximately linearly, although the proportion of these which were preprints fell: By the end of October, over 125,000 publications on COVID-19 had appeared (30,260 preprints; approximately 25%). Given an output of approximately 5 million journal articles and preprints in the entirety of 2020 (according to data from Dimensions; https://dimensions.ai), the publication response to COVID-19 represented >2.5% of outputs during our analysis period. In comparison to other recent outbreaks of global significance caused by emerging RNA viruses, the preprint response to COVID-19 has been much larger; 10,232 COVID-19–related preprints were posted to bioRxiv and medRxiv in the first 10 months of the pandemic; in comparison, only 78 Zika virus–related and 10 Ebola virus–related preprints were posted to bioRxiv during the entire duration of the respective Zika virus epidemic (2015 to 2016) and Western African Ebola virus epidemic (2014 to 2016) (S1A Fig). This surge in COVID-19 preprints is not explained by general increases in preprint server usage; considering counts of outbreak-related and non-outbreak–related preprints for each outbreak (COVID-19, Ebola or Zika virus), preprint type was significantly associated with outbreak (chi-squared, $\chi^2$ = 2559.2, $p$ < 0.001), with the proportion of outbreak-related preprints being greatest for COVID-19.

The 30,260 manuscripts posted as preprints were hosted on a range of preprint servers covering diverse subject areas not limited to biomedical research (Fig 1C, data from [15]). It is important to note that this number includes preprints that may have been posted on multiple preprint servers simultaneously; however, by considering only preprints with unique titles (case insensitive), it appears that this only applies to a small proportion of preprint records (<5%). The total number is preprints is nevertheless likely an underestimation of the true volume of preprints posted, as a number of preprint servers and other repositories (e.g., institutional repositories) that could be expected to host COVID-19 research are not included [15]. Despite being one of the newest preprint servers, medRxiv hosted the largest number of preprints (7,882); the next largest were SSRN (4,180), Research Square (4,089), RePEc (2,774), arXiv (2,592), bioRxiv (2,328), JMIR (1,218), and Preprints.org (1,020); all other preprint servers were found to host <1,000 preprints (Fig 1C).

One of the most frequently cited benefits of preprints is that they allow free access to research findings [16], while a large proportion of journal articles often remain behind subscription paywalls. In response to the pandemic, a number of journal publishers began to alter their open-access policies in relation to COVID-19 manuscripts. One such change was to make COVID-19 literature temporarily open access (at least for the duration of the pandemic), with over 80,000 papers in our dataset being open access (S1B Fig).

## Attributes of COVID-19 preprints posted between January and October 2020

To explore the attributes of COVID-19 preprints in greater detail, we focused our following investigation on two of the most popular preprint servers in the biomedical sciences: bioRxiv and medRxiv. We compared attributes of COVID-19–related preprints posted within our analysis period between January 1 and October 31, 2020 against non-COVID-19–related preprints posted in the same time frame. In total, 44,503 preprints were deposited to bioRxiv and medRxiv in this period, of which the majority (34,271, 77.0%) were non-COVID-19–related

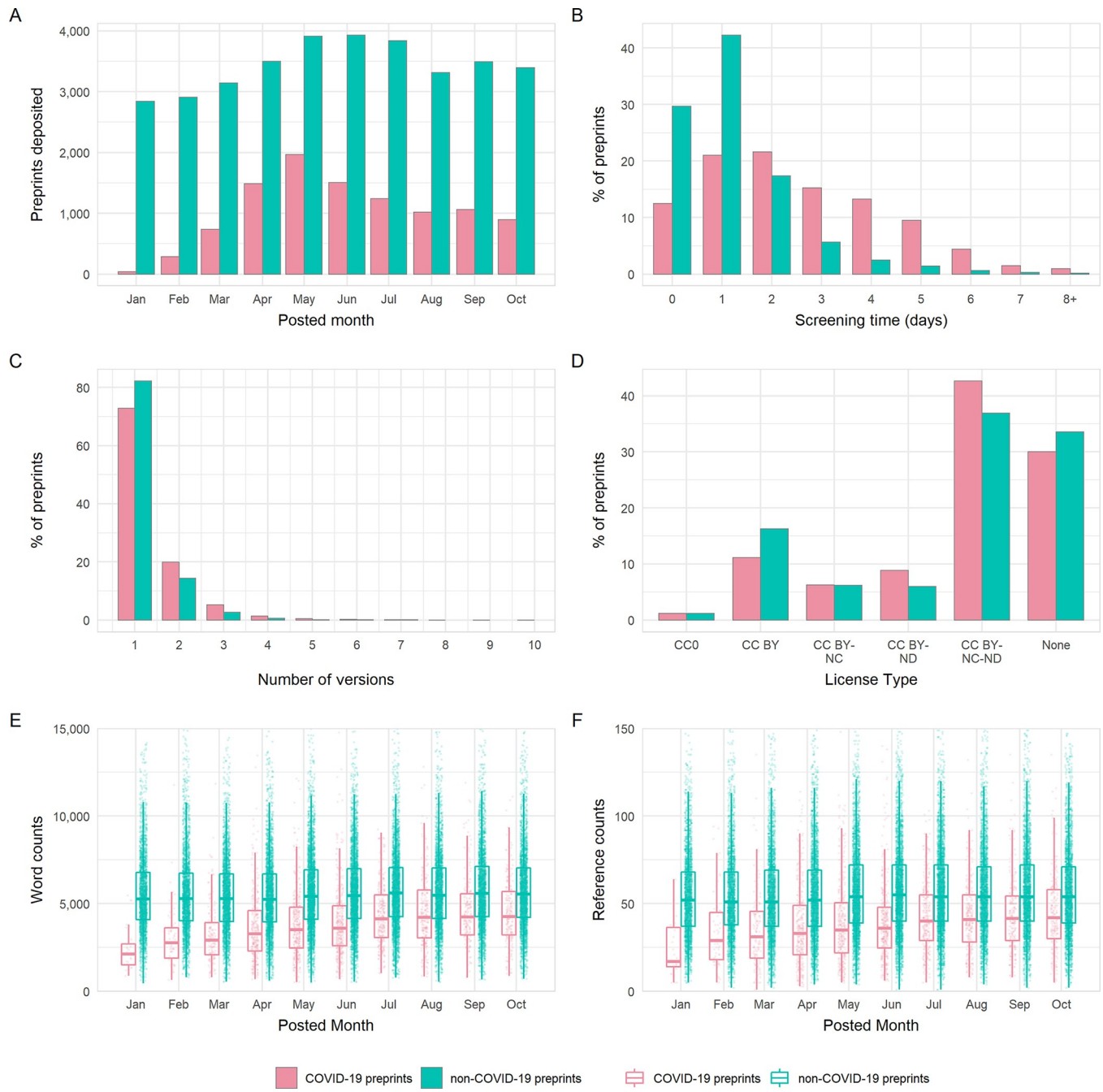

**Fig 2. Comparison of the properties of COVID-19 and non-COVID-19 preprints deposited on bioRxiv and medRxiv between January 1 and October 31, 2020.** (A) Number of new preprints deposited per month. (B) Preprint screening time in days. (C) License type chosen by authors. (D) Number of versions per preprint. (E) Boxplot of preprint word counts, binned by posting month. (F) Boxplot of preprint reference counts, binned by posting month. Boxplot horizontal lines denote lower quartile, median, upper quartile, with whiskers extending to 1.5*IQR. All boxplots additionally show raw data values for individual preprints with added horizontal jitter for visibility. The data underlying this figure may be found in https://github.com/preprinting-a-pandemic/pandemic_preprints and https://zenodo.org/record/4587214#.YEN22Hmnx9A. COVID-19, Coronavirus Disease 2019.

preprints (Fig 2A, S1 Table). During the early phase of the pandemic, the posted monthly volumes of non-COVID-19 preprints was relatively constant, while the monthly volume of COVID-19 preprints increased, peaking at 1,967 in May, and subsequently decreased month

by month. These patterns persisted when the 2 preprint servers were considered independently (S2A Fig). Moreover, COVID-19 preprints have represented the majority of preprints posted to medRxiv each month after February 2020.

The increase in the rate of preprint posting poses challenges for their timely screening. A minor but detectable difference was observed between screening time for COVID-19 and non-COVID-19 preprints (Fig 2B), although this difference appeared to vary with server (2-way ANOVA, interaction term; $F_{1,83333} = 19.22$, $p < 0.001$). Specifically, screening was marginally slower for COVID-19 preprints than for non-COVID-19 preprints deposited to medRxiv (mean difference = 0.16 days; Tukey honest significant difference [HSD] test, $p < 0.001$), but not to bioRxiv ($p = 0.981$). The slower screening time for COVID-19 preprints was a result of more of these preprints being hosted on medRxiv, which had slightly longer screening times overall; bioRxiv screened preprints approximately 2 days quicker than medRxiv independent of COVID-19 status (both $p < 0.001$; S2B Fig, S1 Table).

Preprint servers offer authors the opportunity to post updated versions of a preprint, enabling them to incorporate feedback, correct mistakes, or add additional data and analysis. The majority of preprints existed as only a single version for both COVID-19 and non-COVID-19 works, with very few preprints existing in more than 2 versions (Fig 2C). This may somewhat reflect the relatively short time span of our analysis period. Although distributions were similar, COVID-19 preprints appeared to have a slightly greater number of versions, 1 [IQR 1] versus 1 [IQR 0]; Mann–Whitney test, $p < 0.001$). The choice of preprint server did not appear to impact on the number of versions (S2C Fig, S1 Table).

bioRxiv and medRxiv allow authors to select from a number of different Creative Commons (https://creativecommons.org/) license types when depositing their work: CC0 (No Rights Reserved), CC-BY (Attribution), CC BY-NC (Attribution, Noncommercial), CC-BY-ND (Attribution, No Derivatives), and CC-BY-NC-ND (Attribution, Noncommercial, No Derivatives). Authors may also select to post their work without a license (i.e., All Rights Reserved) that allows text and data mining. A previous analysis has found that bioRxiv authors tend to post preprints under the more restrictive license types [17], although there appears to be some confusion among authors as to the precise implications of each license type [18]. License choice was significantly associated with preprint category (chi-squared, $\chi^2 = 336.0$, df = 5, $p < 0.001$); authors of COVID-19 preprints were more likely to choose the more restrictive CC-BY-NC-ND or CC-BY-ND than those of non-COVID-19 preprints and less likely to choose CC-BY (Fig 2D). Again, the choice of preprint server did not appear to impact on the type of license selected by the authors (S2D Fig).

Given the novelty of the COVID-19 research field and rapid speed at which preprints are being posted, we hypothesised that researchers may be posting preprints in a less mature state, or based on a smaller literature base than for non-COVID preprints. To investigate this, we compared the word counts and reference counts of COVID-19 preprints and non-COVID-19 preprints from bioRxiv (at the time of data extraction, HTML full texts from which word and reference counts were derived were not available for medRxiv) (Fig 2E). We found that COVID-19 preprints are on average 32% shorter in length than non-COVID-19 preprints (median, 3,965 [IQR 2,433] versus 5,427 [IQR 2,790]; Mann–Whitney test, $p < 0.001$) (S1 Table). Although the length of preprints gradually increased over the analysis period, COVID-19 preprints remained shorter than non-COVID-19 preprints with a similar difference in word count, even when adjusted for factors such as authorship team size and bioRxiv subject categorisation (S1 Model, S2 and S3 Tables). COVID-19 preprints also contain fewer references than non-COVID-19 preprints (Fig 2F), although not fewer than expected relative to overall preprint length, as little difference was detected in reference:word count ratios (median, 1:103 versus 1:101; $p = 0.052$). As word counts increased over time, the reference counts per preprint also steadily increased.

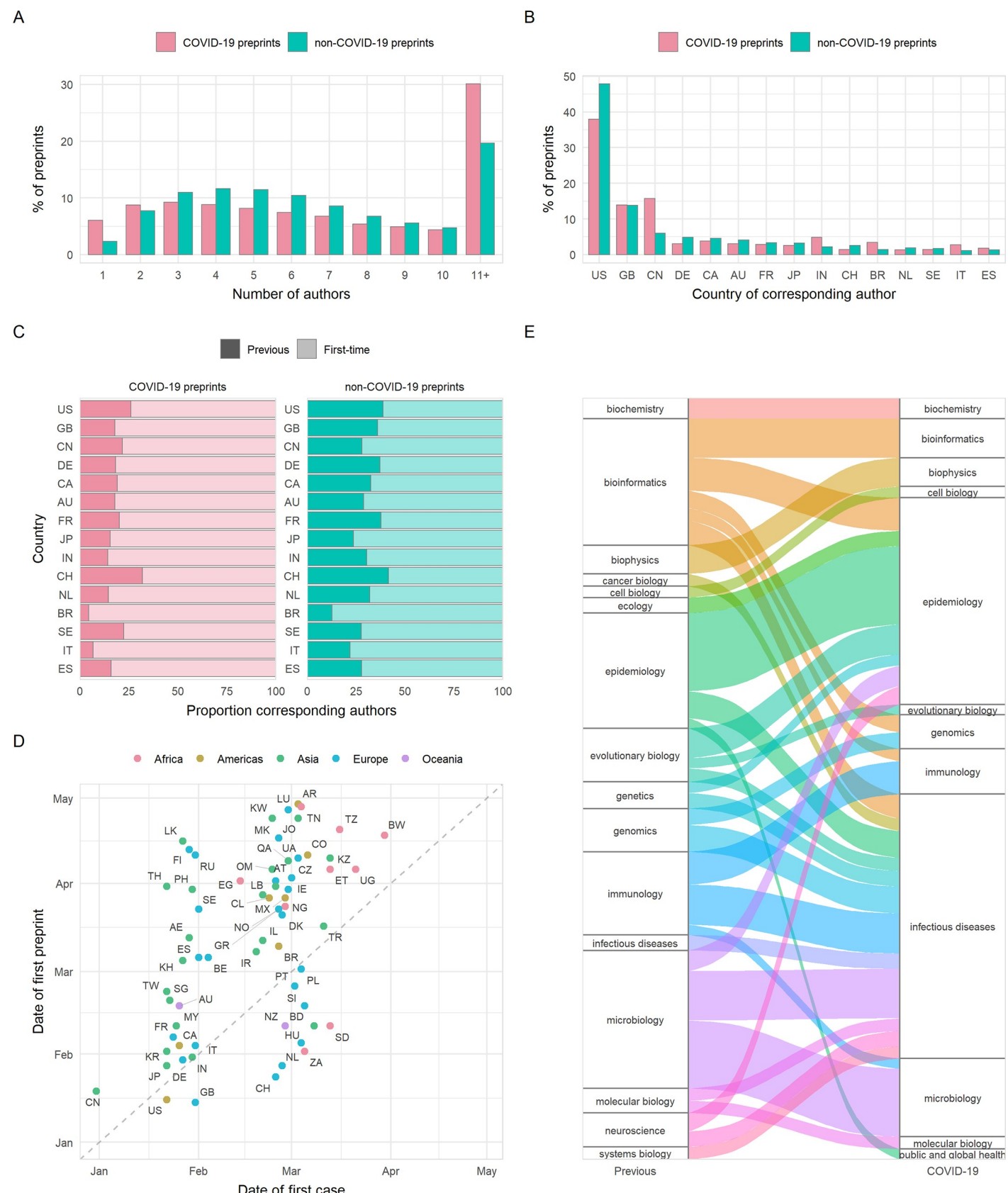

**Fig 3. Properties of authors of COVID-19 and non-COVID-19 preprints deposited on bioRxiv and medRxiv between January 1 and October 31, 2020.** (A) Proportion of preprints with *N* authors. (B) Proportion of preprints deposited by country of corresponding author (top 15 countries by total preprint volume are shown). (C) Proportions of COVID-19 and non-COVID-19 corresponding authors from each of the top 15 countries shown in (B) that had previously posted a preprint (darker bar) or were posting a preprint for the first time (lighter bar). (D) Correlation between date of the first preprint originating from a country (according to the affiliation of the corresponding author) and the date of the first confirmed case from the same country for COVID-19 preprints. (E) Change in bioRxiv/medRxiv preprint posting category for COVID-19 preprint authors compared to their previous preprint (COVID-19 or non-COVID-19), for category combinations with n > = 5 authors. For all panels containing country information, labels refer to ISO 3166 character codes. The data underlying this figure may be found in https://github.com/preprinting-a-pandemic/pandemic_preprints and https://zenodo.org/record/4587214#.YEN22Hmnx9A. COVID-19, Coronavirus Disease 2019.

## Scientists turned to preprints for the first time to share COVID-19 science

The number of authors per preprint may give an additional indication as to the amount of work, resources used, and the extent of collaboration in a manuscript. Although little difference was seen in number of authors between preprint servers (S1 Table), COVID-19 preprints had a marginally higher number of authors than non-COVID-19 preprints on average (median, 7 [IQR 8] versus 6 [IQR 5]; $p < 0.001$), due to the greater likelihood of large (11+) authorship team sizes (Fig 3A). However, single-author preprints were approximately 2.6 times more common for COVID-19 (6.1% of preprints) than non-COVID-19 preprints (2.3% of preprints) (Fig 3A).

The largest proportion of preprints in our dataset were from corresponding authors in the United States, followed by significant proportions from the United Kingdom and China (Fig 3B). It is notable that China is overrepresented in terms of COVID-19 preprints compared to its non-COVID-19 preprint output: 39% of preprints from Chinese corresponding authors were COVID-19 related, compared to 16.5% of the US output and 20.1% of the UK output. We also found a significant association for corresponding authors between preprint type (COVID-19 or non-COVID-19) and whether this was the author's first bioRxiv or medRxiv preprint (chi-squared, $\chi^2 = 840.4$, df = 1, $p < 0.001$). Among COVID-19 corresponding authors, 85% were posting a preprint for the first time, compared to 69% of non-COVID-19 corresponding authors in the same period. To further understand which authors have been drawn to begin using preprints since the pandemic began, we stratified these groups by country (S4 Table) and found significant associations for the US, UK, Germany, India (Bonferroni adjusted $p < 0.001$), France, Canada, Italy ($p < 0.01$), and China ($p < 0.05$). In all cases, a higher proportion were posting a preprint for the first time among COVID-19 corresponding authors than non-COVID-19 corresponding authors. Moreover, we found that most countries posted their first COVID-19 preprint close to the time of their first confirmed COVID-19 case (Fig 3D), with weak positive correlation considering calendar days of both events (Spearman rank; $\rho = 0.54$, $p < 0.001$). Countries posting a COVID-19 preprint in advance of their first confirmed case were mostly higher-income countries (e.g., US, UK, New Zealand, and Switzerland). COVID-19 preprints were deposited from over 100 countries, highlighting the global response to the pandemic.

There has been much discussion regarding the appropriateness of researchers switching to COVID-19 research from other fields [19]. To quantify whether this phenomenon was detectable within the preprint literature, we compared the bioRxiv or medRxiv category of each COVID-19 preprint to the most recent previous non-COVID-19 preprint (if any) from the same corresponding author. Most corresponding authors were not drastically changing fields, with category differences generally spanning reasonably related areas. For example, some authors that previously posted preprints in evolutionary biology have posted COVID-19 preprints in microbiology (Fig 3E). This suggests that—at least within the life sciences—principal investigators are utilising their labs' skills and resources in an expected manner in their contributions to COVID-19 research.

## COVID-19 preprints were published quicker than non-COVID-19 preprints

Critics have previously raised concerns that by forgoing the traditional peer-review process, preprint servers could be flooded by poor-quality research [20,21]. Nonetheless, earlier analyses have shown that a large proportion of preprints (approximately 70%) in the biomedical sciences are eventually published in peer-reviewed scientific journals [8]. We assessed differences in publication outcomes for COVID-19 versus non-COVID-19 preprints during our analysis period, which may be partially related to differences in preprint quality. Published status (published/unpublished) was significantly associated with preprint type (chi-squared, $\chi^2 = 186.2$, df = 1, $p < 0.001$); within our time frame, 21.1% of COVID-19 preprints were published in total by the end of October, compared to 15.4% of non-COVID preprints. As expected, greater proportions published were seen among preprints posted earlier, with over 40% of COVID-19 preprints submitted in January published by the end of October and less than 10% for those published in August or later (Fig 4A). Published COVID-19 preprints were distributed across

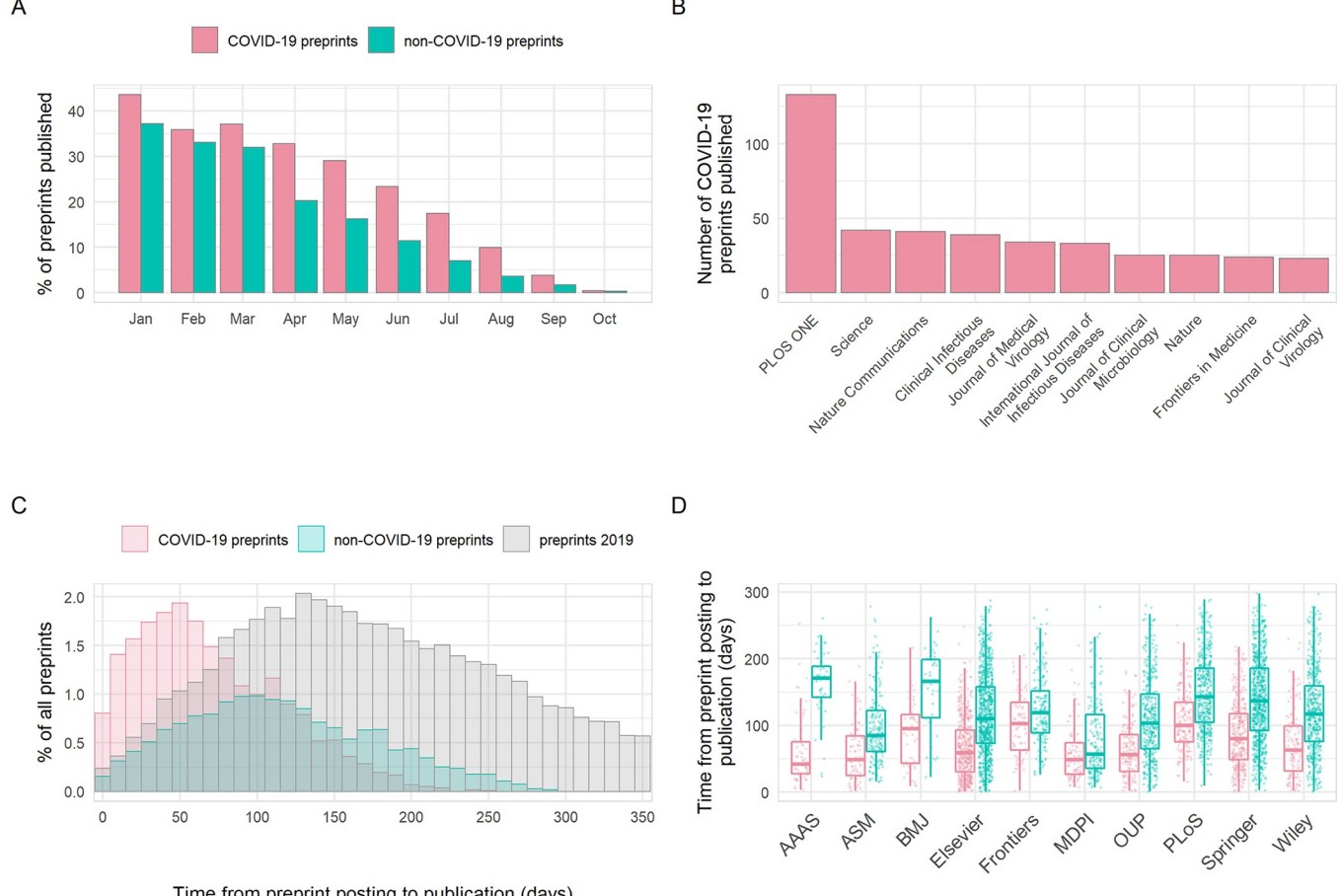

**Fig 4. Publication outcomes of COVID-19 and non-COVID-19 preprints deposited on bioRxiv and medRxiv between January 1 and October 31, 2020.** (A) Percentage of COVID-19 versus non-COVID-19 preprints published in peer-reviewed journals, by preprint posting month. (B) Destination journals for COVID-19 preprints that were published within our analysis period. Shown are the top 10 journals by publication volume. (C) Distribution of the number of days between posting a preprint and subsequent journal publication for COVID-19 preprints (red), non-COVID-19 preprints posted during the same period (January to October 2020) (green), and non-COVID-19 preprints posted between January and December 2019 (grey). (D) Time from posting on bioRxiv or medRxiv to publication categorised by publisher. Shown are the top 10 publishers by publication volume. Boxplot horizontal lines denote lower quartile, median, upper quartile, with whiskers extending to 1.5*IQR. All boxplots additionally show raw data values for individual preprints with added horizontal jitter for visibility. The data underlying this figure may be found in https://github.com/preprinting-a-pandemic/pandemic_preprints and https://zenodo.org/record/4587214#.YEN22Hmnx9A. COVID-19, Coronavirus Disease 2019.

many journals, with clinical or multidisciplinary journals tending to publish the most COVID-19 preprints (Fig 4B). To determine how publishers were prioritising COVID-19 research, we compared the time from preprint posting to publication in a journal. The time interval from posting to subsequent publication was significantly reduced for COVID-19 preprints by a difference in medians of 48 days compared to non-COVID-19 preprints posted in the same time period (68 days [IQR 69] versus 116 days [IQR 90]; Mann–Whitney test, $p < 0.001$). This did not appear to be driven by any temporal changes in publishing practices, as the distribution of publication times for non-COVID-19 preprints was similar to our control time frame of January to December 2019 (Fig 4C). This acceleration additionally varied between publishers (2-way ANOVA, interaction term preprint type*publisher; $F_{9,5273} = 6.58$, $p < 0.001$) and was greatest for the American Association for the Advancement of Science (AAAS) at an average difference of 102 days (Tukey HSD; $p < 0.001$) (Fig 4D).

## Extensive access of preprint servers for COVID-19 research

At the start of our time window, COVID-19 preprints received abstract views at a rate over 18 times that of non-COVID-19 preprints (Fig 5A) (time-adjusted negative binomial regression; rate ratio = 18.2, z = 125.0, $p < 0.001$) and downloads at a rate of almost 30 times (Fig 5B) (rate ratio = 27.1, z = 124.2, $p < 0.001$). Preprints posted later displayed lower usage rates, in part due to the reduced length of time they were online and able to accrue views and downloads. However, decreases in both views and downloads by posting date was stronger for COVID-19 preprints versus non-COVID-19 preprints (preprint type*calendar day interaction terms, both $p < 0.001$); each additional calendar month in posting date resulted in an estimated 24.3%/7.4% reduction in rate of views and an estimated 28.5%/12.0% reduction in rate of downloads for COVID-19/non-COVID-19 preprints, respectively. Similar trends of decrease were observed when restricting view and download data to the first respective month of each preprint, with highest rates of usage for those posted in January (S3A and S3B Fig). The disparity between COVID-19 and non-COVID-19 preprints suggests that either COVID-19 preprints continued to slowly accumulate total usage well beyond their first month online (Fig 5) and/or they received a more diluted share of relative initial interest as larger volumes of preprints (and publications) were available by later months (Fig 1B).

To confirm that usage of COVID-19 and non-COVID-19 preprints was not an artefact of differing preprint server reliance during the pandemic, we compared usage rates during the pandemic period with those from the previous year (January to December 2019), as a non-pandemic control period. Beyond the expected effect of fewer views/downloads of preprints that have been uploaded for a shorter time, the usage data did not differ from that prior to the pandemic (S3C and S3D Fig).

Secondly, we investigated usage across additional preprint servers (data kindly provided by each of the server operators). We found that COVID-19 preprints were consistently downloaded more than non-COVID-19 preprints during our time frame, regardless of which preprint server hosted the manuscript (S3E Fig), although the gap in downloads varied between server (2-way ANOVA, interaction term; $F_{3,89990} = 126.6$, $p < 0.001$). Server usage differences were more pronounced for COVID-19 preprints; multiple post hoc comparisons confirmed that bioRxiv and medRxiv received significantly higher usage per COVID-19 preprint than all other servers for which data were available (Tukey HSD; all p values $< 0.001$). However, for non-COVID-19 preprints, the only observed pairwise differences between servers indicated greater bioRxiv and medRxiv usage than Research Square (Tukey HSD; $p < 0.001$). This suggests that specific attention has been given disproportionately to bioRxiv and medRxiv as repositories for COVID-19 research.

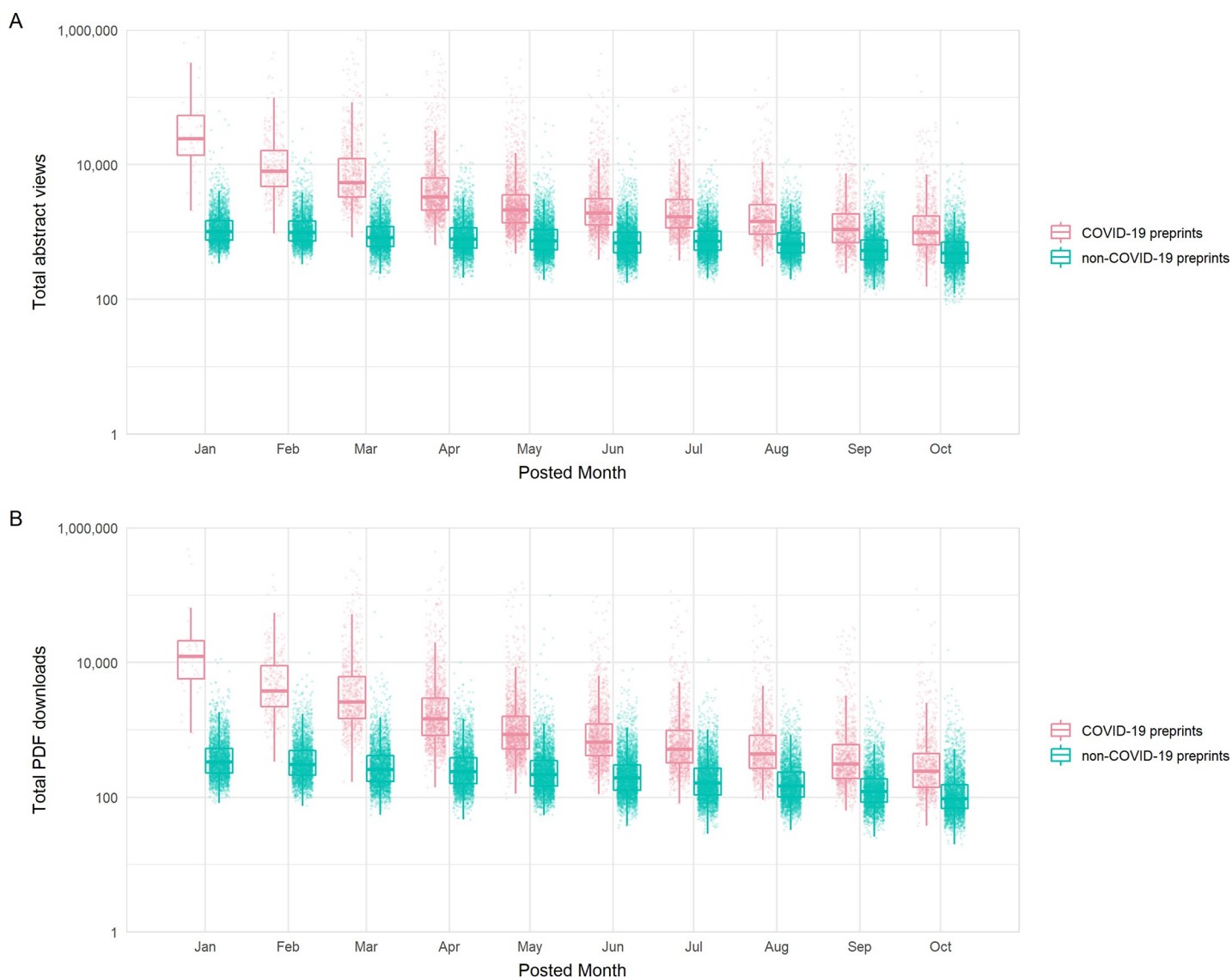

**Fig 5. Access statistics for COVID-19 and non-COVID-19 preprints posted on bioRxiv and medRxiv.** (A) Boxplots of abstract views, binned by preprint posting month. (B) Boxplots of PDF downloads, binned by preprint posting month. Boxplot horizontal lines denote lower quartile, median, upper quartile, with whiskers extending to 1.5*IQR. All boxplots additionally show raw data values for individual preprints with added horizontal jitter for visibility. The data underlying this figure may be found in https://github.com/preprinting-a-pandemic/pandemic_preprints and https://zenodo.org/record/4587214#.YEN22Hmnx9A. COVID-19, Coronavirus Disease 2019.

## COVID-19 preprints were shared and cited more widely than non-COVID-19 preprints

We quantified the citation and online sharing behaviour of COVID-19 preprints using citation count data from Dimensions (https://dimensions.ai) and counts of various altmetric indicators using data from Altmetric (https://altmetric.com) (Fig 6; further details on data sources in Methods section). In terms of citations, we found higher proportions overall of COVID-19 preprints that received at least a single citation (57.9%) than non-COVID-19 preprints (21.5%) during our study period of January 1 to October 31, 2020, although the citation coverage expectedly decreased for both groups for newer posted preprints (Fig 6A). COVID-19 pre-prints also have greater total citation counts than non-COVID-19 preprints (time-adjusted

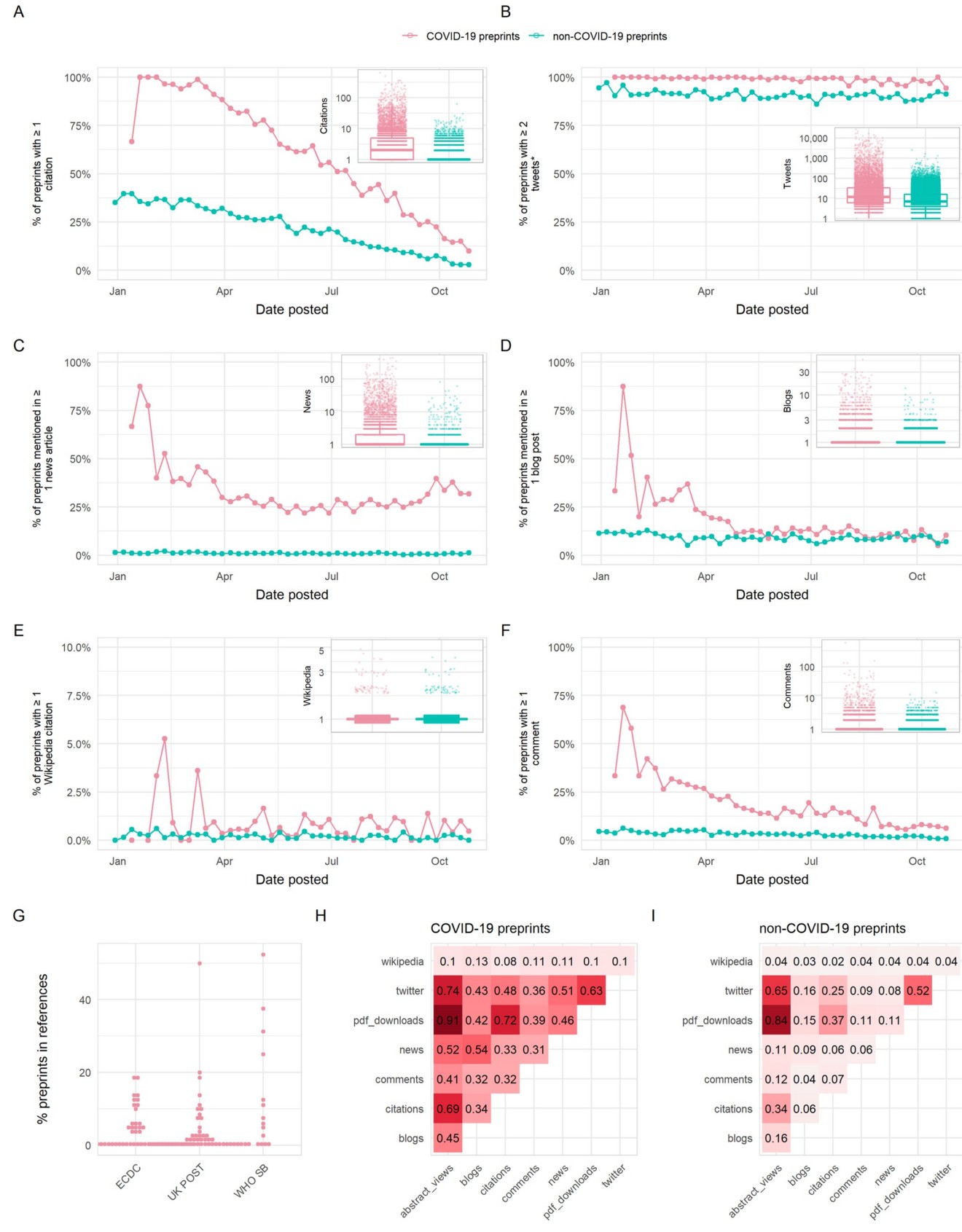

**Fig 6. Usage of COVID-19 and non-COVID-19 preprints posted on bioRxiv and medRxiv between January 1 and October 31, 2020.** Panels (A)–(F) show the proportion of preprints receiving at least 1 citation or mention in a given source, with the exception of panel (B) which shows the proportion of preprints receiving at least 2 tweets (to account for the fact that each preprint is tweeted once automatically by the official bioRxiv/medRxiv Twitter accounts). The inset in each panel shows a boxplot comparing citations/mentions for all COVID-19 and non-COVID-19 preprints posted within our analysis period. Boxplot horizontal lines denote lower quartile, median, upper quartile, with whiskers extending to 1.5*IQR. All boxplots additionally show raw data values for individual preprints with added horizontal jitter for visibility. Data are plotted on a log-scale with +1 added to each count for visualisation. (G) Proportion of preprints included in reference lists of policy documents from 3 sources: the ECDC, UK POST, and WHO SB. (H) Spearman correlation matrix between indicators shown in panels (A)–(F), as well as abstract views and PDF downloads for COVID-19 preprints. (I) Spearman correlation matrix between indicators shown in panels (A)–(F), in addition to abstract views and PDF downloads for non-COVID-19 preprints. The data underlying this figure may be found in https://github.com/preprinting-a-pandemic/pandemic_preprints and https://zenodo.org/record/4587214#.YEN22Hmnx9A. COVID-19, Coronavirus Disease 2019; ECDC, European Centre for Disease Prevention and Control; UK POST, United Kingdom Parliamentary Office of Science and Technology; WHO SB, World Health Organization Scientific Briefs.

negative binomial regression; rate ratio = 13.7, z = 116.3, $p < 0.001$). The highest cited COVID-19 preprint had 652 citations, with the 10th most cited COVID-19 preprint receiving 277 citations (Table 1); many of the highest cited preprints focussed on the viral cell receptor, angiotensin converting enzyme 2 (ACE2), or the epidemiology of COVID-19.

Sharing of preprints on Twitter may provide an indicator of the exposure of wider public audiences to preprints. COVID-19 preprints received greater Twitter coverage (98.9% received >1 tweet) than non-COVID-19 preprints (90.7%) (note that the threshold for Twitter coverage was set at 1 rather than 0, to account for automated tweets by the official bioRxiv and medRxiv Twitter accounts) and were tweeted at an overall greater rate than non-COVID-19 preprints (rate ratio = 7.6, z = 135.7, $p < 0.001$) (Fig 6B). The most tweeted non-COVID-19 preprint received 1,656 tweets, whereas 8 of the top 10 tweeted COVID-19 preprints were tweeted over 10,500 times each (Table 2). Many of the top 10 tweeted COVID-19 preprints were related to transmission, reinfection, or seroprevalence. The most tweeted COVID-19 preprint (26,763 tweets) was a study investigating antibody seroprevalence in California [22]. The fourth most tweeted COVID-19 preprint was a widely criticised (and later withdrawn) study linking the SARS-CoV-2 spike protein to HIV-1 glycoproteins [23].

**Table 1. Top 10 cited COVID-19 preprints.**

| Rank | Source | DOI | Title | Posted date | Citations |
|------|--------|-----|-------|-------------|-----------|
| 1 | medRxiv | 10.1101/2020.02.06.20020974 | Clinical characteristics of 2019 novel coronavirus infection in China | February 9, 2020 | 652 |
| 2 | bioRxiv | 10.1101/2020.02.07.937862 | Severe acute respiratory syndrome-related coronavirus—The species and its viruses, a statement of the Coronavirus Study Group | February 11, 2020 | 513 |
| 3 | medRxiv | 10.1101/2020.01.23.20018549 | Novel coronavirus 2019-nCoV: early estimation of epidemiological parameters and epidemic predictions | January 24, 2020 | 361 |
| 4 | bioRxiv | 10.1101/2020.01.26.919985 | Single-cell RNA expression profiling of ACE2, the putative receptor of Wuhan 2019-nCov | January 26, 2020 | 359 |
| 5 | medRxiv | 10.1101/2020.03.22.20040758 | Efficacy of hydroxychloroquine in patients with COVID-19: results of a randomized clinical trial | March 30, 2020 | 358 |
| 6 | bioRxiv | 10.1101/2020.01.31.929042 | The novel coronavirus 2019 (2019-nCoV) uses the SARS-coronavirus receptor ACE2 and the cellular protease TMPRSS2 for entry into target cells | January 31, 2020 | 313 |
| 7 | bioRxiv | 10.1101/2020.02.03.931766 | Specific ACE2 Expression in Cholangiocytes May Cause Liver Damage After 2019-nCoV Infection | February 4, 2020 | 304 |
| 8 | bioRxiv | 10.1101/2020.01.22.914952 | Discovery of a novel coronavirus associated with the recent pneumonia outbreak in humans and its potential bat origin | January 23, 2020 | 302 |
| 9 | bioRxiv | 10.1101/2020.01.30.927806 | The digestive system is a potential route of 2019-nCov infection: a bioinformatics analysis based on single-cell transcriptomes | January 31, 2020 | 285 |
| 10 | medRxiv | 10.1101/2020.02.11.20021493 | Laboratory diagnosis and monitoring the viral shedding of 2019-nCoV infections | February 12, 2020 | 277 |

COVID-19, Coronavirus Disease 2019.

**Table 2. Top 10 tweeted COVID-19 preprints.**

| Rank | Source | DOI | Title | Posted date | Tweets | News articles | Blogs |
|------|--------|-----|-------|-------------|--------|---------------|-------|
| 1 | medRxiv | 10.1101/ 2020.04.14.20062463 | COVID-19 Antibody Seroprevalence in Santa Clara County, California | April 17, 2020 | 26,763 | 434 | 55 |
| 2 | medRxiv | 10.1101/ 2020.04.04.20053058 | Indoor transmission of SARS-CoV-2 | April 7, 2020 | 21,831 | 187 | 34 |
| 3 | medRxiv | 10.1101/ 2020.07.15.20151852 | Effect of Hydroxychloroquine in Hospitalized Patients with COVID-19: Preliminary results from a multi-centre, randomized, controlled trial. | July 15, 2020 | 17,534 | 83 | 5 |
| 4 | bioRxiv | 10.1101/ 2020.01.30.927871 | Uncanny similarity of unique inserts in the 2019-nCoV spike protein to HIV-1 gp120 and Gag | January 31, 2020 | 16,608 | 102 | 25 |
| 5 | medRxiv | 10.1101/ 2020.05.19.20105999 | SARS-CoV-2 RNA concentrations in primary municipal sewage sludge as a leading indicator of COVID-19 outbreak dynamics | May 22, 2020 | 16,582 | 63 | 8 |
| 6 | medRxiv | 10.1101/ 2020.03.22.20040758 | Efficacy of hydroxychloroquine in patients with COVID-19: results of a randomized clinical trial | March 30, 2020 | 14,614 | 106 | 18 |
| 7 | medRxiv | 10.1101/ 2020.10.14.20212555 | Multi-organ impairment in low-risk individuals with long COVID | October 16, 2020 | 12,871 | 34 | 6 |
| 8 | medRxiv | 10.1101/ 2020.03.09.20033217 | Aerosol and surface stability of HCoV-19 (SARS-CoV-2) compared to SARS-CoV-1 | March 10, 2020 | 12,484 | 354 | 29 |
| 9 | medRxiv | 10.1101/ 2020.08.03.20167395 | Viable SARS-CoV-2 in the air of a hospital room with COVID-19 patients | August 4, 2020 | 11,770 | 121 | 8 |
| 10 | medRxiv | 10.1101/ 2020.03.30.20048165 | Association of BCG vaccination policy with prevalence and mortality of COVID-19 | April 6, 2020 | 10,701 | 7 | 0 |

COVID-19, Coronavirus Disease 2019.

To better understand the discussion topics associated with highly tweeted preprints, we analysed the hashtags used in original tweets (i.e., excluding retweets) mentioning the top 100 most tweeted COVID-19 preprints (S4A Fig). In total, we collected 30,213 original tweets containing 11,789 hashtags; we filtered these hashtags for those occurring more than 5 times and removed a selection of generic or overused hashtags directly referring to the virus (e.g., "#coronavirus" and "#covid-19"), leaving a final set of 2,981 unique hashtags. While many of the top-used hashtags were direct, neutral references to the disease outbreak such as "#coronavirusoutbreak" and "#wuhan," we also found a large proportion of politicised tweets using hashtags associated with conspirational ideologies (e.g., "#qanon," "#wwg1wga," an abbreviation of "Where We Go One, We Go All" a tag commonly used by QAnon supporters), xenophobia (e.g., "#chinazi"), or US-specific right-wing populism (e.g., "#maga"). Other hashtags also referred to topics directly associated with controversial preprints, e.g., "#hydroxychloroquine" and "#hiv," both of which were major controversial topics associated with several of the top 10 most tweeted preprints.

As well as featuring heavily on social media, COVID-19 research has also pervaded print and online news media. In terms of coverage, 28.7% of COVID-19 preprints were featured in at least a single news article, compared to 1.0% of non-COVID-19 preprints (Fig 6C), and were used overall in news articles at a rate almost 100 times that of non-COVID-19 preprints (rate ratio = 92.8, z = 83.3, $p < 0.001$). The top non-COVID-19 preprint was reported in 113 news articles, whereas the top COVID-19 preprints were reported in over 400 news articles (Table 3). Similarly, COVID-19 preprints were also used more in blogs (coverage COVID-19/ non-COVID-19 preprints = 14.3%/9.1%, rate ratio = 3.73, z = 37.3, $p < 0.001$) and Wikipedia articles (coverage COVID-19/non-COVID-19 preprints = 0.7%/0.2%, rate ratio = 4.47, z = 7.893, $p < 0.001$) at significantly greater rates than non-COVID-19 preprints (Fig 6D and 6E, Table 4). We noted that several of the most widely disseminated preprints that we

**Table 3. Top 10 COVID-19 preprints covered by news organisations.**

| Rank | Source | DOI | Title | Posted date | Tweets | News articles | Blogs |
|---|---|---|---|---|---|---|---|
| 1 | bioRxiv | 10.1101/ 2020.04.29.069054 | Spike mutation pipeline reveals the emergence of a more transmissible form of SARS-CoV-2 | April 30, 2020 | 6,848 | 449 | 29 |
| 2 | medRxiv | 10.1101/ 2020.04.14.20062463 | COVID-19 Antibody Seroprevalence in Santa Clara County, California | April 17, 2020 | 26,763 | 434 | 55 |
| 3 | medRxiv | 10.1101/ 2020.04.16.20065920 | Outcomes of hydroxychloroquine usage in United States veterans hospitalized with Covid-19 | April 21, 2020 | 10,385 | 411 | 27 |
| 4 | medRxiv | 10.1101/ 2020.10.15.20209817 | Repurposed antiviral drugs for COVID-19; interim WHO SOLIDARITY trial results | October 15, 2020 | 8,569 | 396 | 25 |
| 5 | medRxiv | 10.1101/ 2020.03.09.20033217 | Aerosol and surface stability of HCoV-19 (SARS-CoV-2) compared to SARS-CoV-1 | March 10, 2020 | 12,484 | 354 | 29 |
| 6 | medRxiv | 10.1101/ 2020.05.15.20103655 | Differential Effects of Intervention Timing on COVID-19 Spread in the United States | May 20, 2020 | 1,831 | 295 | 16 |
| 7 | medRxiv | 10.1101/ 2020.07.09.20148429 | Longitudinal evaluation and decline of antibody responses in SARS-CoV-2 infection | July 11, 2020 | 2,167 | 281 | 27 |
| 8 | medRxiv | 10.1101/ 2020.08.12.20169359 | Effect of Convalescent Plasma on Mortality among Hospitalized Patients with COVID-19: Initial Three-Month Experience | August 12, 2020 | 2,746 | 264 | 26 |
| 9 | medRxiv | 10.1101/ 2020.06.22.20137273 | Effect of Dexamethasone in Hospitalized Patients with COVID-19: Preliminary Report | June 22, 2020 | 5,698 | 246 | 26 |
| 10 | medRxiv | 10.1101/ 2020.03.11.20031096 | Relationship between the ABO Blood Group and the COVID-19 Susceptibility | March 16, 2020 | 4,055 | 245 | 23 |

COVID-19, Coronavirus Disease 2019.

**Table 4. Top 10 commented on COVID-19 preprints.**

| Rank | Source | DOI | Title | Posted date | Comments count |
|---|---|---|---|---|---|
| 1 | medRxiv | 10.1101/ 2020.04.14.20062463 | COVID-19 Antibody Seroprevalence in Santa Clara County, California | April 17, 2020 | 582 |
| 2 | medRxiv | 10.1101/ 2020.03.24.20042937 | Correlation between universal BCG vaccination policy and reduced morbidity and mortality for COVID-19: an epidemiological study | March 28, 2020 | 149 |
| 3 | bioRxiv | 10.1101/ 2020.01.30.927871 | Uncanny similarity of unique inserts in the 2019-nCoV spike protein to HIV-1 gp120 and Gag | January 31, 2020 | 129 |
| 4 | medRxiv | 10.1101/ 2020.04.16.20065920 | Outcomes of hydroxychloroquine usage in United States veterans hospitalized with Covid-19 | April 21, 2020 | 129 |
| 5 | bioRxiv | 10.1101/ 2020.04.29.069054 | Spike mutation pipeline reveals the emergence of a more transmissible form of SARS-CoV-2 | April 30, 2020 | 75 |
| 6 | medRxiv | 10.1101/ 2020.03.11.20031096 | Relationship between the ABO Blood Group and the COVID-19 Susceptibility | March 16, 2020 | 72 |
| 7 | medRxiv | 10.1101/ 2020.03.27.20043752 | Forecasting COVID-19 impact on hospital bed-days, ICU-days, ventilator-days and deaths by US state in the next 4 months | March 30, 2020 | 61 |
| 8 | medRxiv | 10.1101/ 2020.03.22.20040758 | Efficacy of hydroxychloroquine in patients with COVID-19: results of a randomized clinical trial | March 30, 2020 | 58 |
| 9 | medRxiv | 10.1101/ 2020.04.16.20067835 | Saliva is more sensitive for SARS-CoV-2 detection in COVID-19 patients than nasopharyngeal swabs | April 22, 2020 | 56 |
| 10 | medRxiv | 10.1101/ 2020.04.05.20054361 | Population-level COVID-19 mortality risk for non-elderly individuals overall and for non-elderly individuals without underlying diseases in pandemic epicenters | April 8, 2020 | 53 |

COVID-19, Coronavirus Disease 2019.

**Table 5. Top 10 most blogged COVID-19 preprints.**

| Rank | Source | DOI | Title | Posted date | Tweets | News articles | Blogs |
|---|---|---|---|---|---|---|---|
| 1 | medRxiv | 10.1101/ 2020.04.14.20062463 | COVID-19 Antibody Seroprevalence in Santa Clara County, California | April 17, 2020 | 26,763 | 434 | 55 |
| 2 | medRxiv | 10.1101/ 2020.04.04.20053058 | Indoor transmission of SARS-CoV-2 | April 7, 2020 | 21,831 | 187 | 34 |
| 3 | bioRxiv | 10.1101/ 2020.04.29.069054 | Spike mutation pipeline reveals the emergence of a more transmissible form of SARS-CoV-2 | April 30, 2020 | 6,848 | 449 | 29 |
| 4 | medRxiv | 10.1101/ 2020.03.09.20033217 | Aerosol and surface stability of HCoV-19 (SARS-CoV-2) compared to SARS-CoV-1 | March 10, 2020 | 12,484 | 354 | 29 |
| 5 | medRxiv | 10.1101/ 2020.04.16.20065920 | Outcomes of hydroxychloroquine usage in United States veterans hospitalized with Covid-19 | April 21, 2020 | 10,385 | 411 | 27 |
| 6 | medRxiv | 10.1101/ 2020.07.09.20148429 | Longitudinal evaluation and decline of antibody responses in SARS-CoV-2 infection | July 11, 2020 | 2,167 | 281 | 27 |
| 7 | medRxiv | 10.1101/ 2020.06.22.20137273 | Effect of Dexamethasone in Hospitalized Patients with COVID-19: Preliminary Report | June 22, 2020 | 5,698 | 246 | 26 |
| 8 | medRxiv | 10.1101/ 2020.08.12.20169359 | Effect of Convalescent Plasma on Mortality among Hospitalized Patients with COVID-19: Initial Three-Month Experience | August 12, 2020 | 2,746 | 264 | 26 |
| 9 | bioRxiv | 10.1101/ 2020.01.30.927871 | Uncanny similarity of unique inserts in the 2019-nCoV spike protein to HIV-1 gp120 and Gag | January 31, 2020 | 16,608 | 102 | 25 |
| 10 | bioRxiv | 10.1101/ 2020.03.30.015347 | Susceptibility of ferrets, cats, dogs, and different domestic animals to SARS-coronavirus-2 | March 31, 2020 | 4,168 | 209 | 25 |
| 11 | medRxiv | 10.1101/ 2020.10.15.20209817 | Repurposed antiviral drugs for COVID-19; interim WHO SOLIDARITY trial results | October 15, 2020 | 8,569 | 396 | 25 |

COVID-19, Coronavirus Disease 2019.

classified as being non-COVID-19 related featured topics nonetheless relevant to generalised infectious disease research, such as human respiratory physiology and personal protective equipment.

A potential benefit of preprints is that they allow authors to receive an incorporate feedback from the wider community prior to journal publication. To investigate feedback and engagement with preprints, we quantified the number of comments received by preprints directly via the commenting system on the bioRxiv and medRxiv platforms. We found that non-COVID-19 preprints were commented upon less frequently compared to COVID-19 preprints (coverage COVID-19/non-COVID-19 preprints = 15.9%/3.1%, time-adjusted negative binomial regression; rate ratio = 11.0, z = 46.5, $p < 0.001$) (Fig 6F); the most commented non-COVID-19 preprint received only 68 comments, whereas the most commented COVID-19 preprint had over 580 comments (Table 5). One preprint, which had 129 comments, was retracted within 3 days of being posted following intense public scrutiny (Table 4, doi: 10.1101/2020.01.30.927871). As the pandemic has progressed, fewer preprints were commented upon. Collectively, these data suggest that the most discussed or controversial COVID-19 preprints are rapidly and publicly scrutinised, with commenting systems being used for direct feedback and discussion of preprints.

Within a set of 81 COVID-19 policy documents (which were manually retrieved from the European Centre for Disease Prevention and Control (ECDC), United Kingdom Parliamentary Office of Science and Technology (UK POST), and World Health Organization Scientific Briefs (WHO SB)), 52 documents cited preprints (Fig 6G). However, these citations occurred at a relatively low frequency, typically constituting less than 20% of the total citations in these 52 documents. Among 255 instances of citation to a preprint, medRxiv was the dominant server cited (n = 209, 82%), with bioRxiv receiving a small number of citations (n = 21) and 5

other servers receiving ≤10 citations each (arXiv, OSF, preprints.org, Research Square, and SSRN). In comparison, only 16 instances of citations to preprints were observed among 38 manually collected non-COVID-19 policy documents from the same sources.

To understand how different usage and sharing indicators may represent the behaviour of different user groups, we calculated the Spearman correlation between the indicators presented above (citations, tweets, news articles, blog mentions, Wikipedia citations, and comment counts) as well as with abstract views and download counts as previously presented (Fig 6H and 6I). Overall, we found stronger correlations between all indicators for COVID-19 preprints compared to non-COVID-19 preprints. For COVID-19 preprints, we found expectedly strong correlation between abstract views and PDF downloads (Spearman $\rho = 0.91$, $p < 0.001$), weak to moderate correlation between the numbers of citations and Twitter shares (Spearman $\rho = 0.48$, $p < 0.001$), and the numbers of citations and news articles (Spearman $\rho = 0.33$, $p < 0.001$) suggesting that the preprints cited extensively within the scientific literature did not necessarily correlate with those that were mostly shared by the wider public on online platforms. There was a slightly stronger correlation between COVID-19 preprints that were most blogged and those receiving the most attention in the news (Spearman $\rho = 0.54$, $p < 0.001$) and moderate correlation between COVID-19 preprints that were most tweeted and those receiving the most attention in the news (Spearman $\rho = 0.51$, $p < 0.001$), suggesting similarity between preprints shared on social media and in news media. Finally, there was a weak correlation between the number of tweets and number of comments received by COVID-19 preprints (Spearman $\rho = 0.36$, $p < 0.001$). Taking the top 10 COVID-19 preprints by each indicator, there was substantial overlap between all indicators except citations (S4B Fig).

In summary, our data reveal that COVID-19 preprints received a significant amount of attention from scientists, news organizations, the general public, and policy-making bodies, representing a departure for how preprints are normally shared (considering observed patterns for non-COVID-19 preprints).

## Discussion

The usage of preprint servers within the biological sciences has been rising since the inception of bioRxiv and other platforms [10,25]. The urgent threat of a global pandemic has catapulted the use of preprint servers as a means of quickly disseminating scientific findings into the public sphere, supported by funding bodies encouraging preprinting for COVID-19 research [26,27]. Our results show that preprints have been widely adopted for the dissemination and communication of COVID-19 research, and in turn, the pandemic has greatly impacted the preprint and science publishing landscape [28].

Changing attitudes and acceptance within the life sciences to preprint servers may be one reason why COVID-19 research is being shared more readily as preprints compared to previous epidemics. In addition, the need to rapidly communicate findings prior to a lengthy review process might be responsible for this observation (Fig 3). A recent study involving qualitative interviews of multiple research stakeholders found "early and rapid dissemination" to be among the most often cited benefits of preprints [16]. These findings were echoed in a survey of approximately 4,200 bioRxiv users [10] and are underscored by the 6-month median lag between posting of a preprint and subsequent journal publication [8,16]. Such timelines for disseminating findings are clearly incompatible with the lightning-quick progression of a pandemic. An analysis of publication timelines for 14 medical journals has shown that some publishers have taken steps to accelerate their publishing processes for COVID-19 research, reducing the time for the peer-review stage (submission to acceptance) on average by 45 days

and the editing stage (acceptance to publication) by 14 days [29], yet this still falls some way short of the approximately 1 to 3 days screening time for bioRxiv and medRxiv preprints (Fig 2B). This advantage may influence the dynamics of preprint uptake: As researchers in a given field begin to preprint, their colleagues may feel pressure to also preprint in order to avoid being scooped. Further studies on understanding the motivations behind posting preprints, for example, through quantitative and qualitative author surveys, may help funders and other stakeholders that support the usage of preprints to address some of the social barriers for their uptake [30].

One of the primary concerns among authors around posting preprints is premature media coverage [16,31]. Many preprint servers created highly visible collections of COVID-19 work, potentially amplifying its visibility. From mid-March 2020, bioRxiv and medRxiv included a banner to explain that preprints should not be regarded as conclusive and not reported on in the news media as established information [32]. Despite this warning message, COVID-19 preprints have received unprecedented coverage on online media platforms (Fig 6). Indeed, even before this warning message was posted, preprints were receiving significant amounts of attention. Twitter has been a particularly notable outlet for communication of preprints, a finding echoed by a recent study on the spread of the wider (i.e., not limited to preprints) COVID-19 research field on Twitter, which found that COVID-19 research was being widely disseminated and driven largely by academic Twitter users [33,34]. Nonetheless, the relatively weak correlation found between citations and other indicators of online sharing (Fig 6H) suggests that the interests of scientists versus the broader public differ significantly: Of the articles in the top 10 most shared on Twitter, in news articles or on blogs, only one is ranked among the top 10 most cited articles (S4B Fig). Hashtags associated with individual, highly tweeted preprints reveal some emergent themes that suggest communication of certain preprints can also extend well beyond scientific audiences (S4A Fig) [34]. These range from good public health practice ("#washyourhands") to right-wing philosophies (#chinalies), conspiracy theories ("#fakenews" and "#endthelockdown"), and xenophobia ("#chinazi"). Many of the negative hashtags have been perpetuated by public figures such as the former President of America and the right-wing media [35,36]. Following President Trump's diagnosis of COVID-19, one investigation found a wave of anti-Asian sentiment and conspiracy theories across Twitter [37]. This type of misinformation is common to new diseases, and social media platforms have recently released a statement outlining their plans to combat this issue [38]. An even greater adoption of open science principles has recently been suggested as one method to counter the misuse of preprints and peer-reviewed articles [24]; this remains an increasingly important discourse.

The fact that news outlets are reporting extensively on COVID-19 preprints (Fig 6C and 6D) represents a marked change in journalistic practice: Pre-pandemic, bioRxiv preprints received very little coverage in comparison to journal articles [25]. This cultural shift provides an unprecedented opportunity to bridge the scientific and media communities to create a consensus on the reporting of preprints [21,39]. Another marked change was observed in the use of preprints in policy documents (Fig 6G). Preprints were remarkably underrepresented in non-COVID-19 policy documents yet present, albeit at relatively low levels, in COVID-19 policy documents. In a larger dataset, two of the top 10 journals which are being cited in policy documents were found to be preprint servers (medRxiv and SSRN in fifth and eighth position, respectively) [40]. This suggests that preprints are being used to directly influence policymakers and decision-making. We only investigated a limited set of policy documents, largely restricted to Europe; whether this extends more globally remains to be explored [41]. In the near future, we aim to examine the use of preprints in policy in more detail to address these questions.

As most COVID-19-preprints were not yet published, concerns regarding quality will persist [20]. This is partially addressed by prominent scientists using social media platforms such as Twitter to publicly share concerns about poor-quality COVID-19 preprints or to amplify high-quality preprints [42]. The use of Twitter to "peer-review" preprints provides additional public scrutiny of manuscripts that can complement the more opaque and slower traditional peer-review process. In addition to Twitter, the comments section of preprint servers can be used as a public forum for discussion and review. However, an analysis of all bioRxiv comments from September 2019 found a very limited number of peer-review style comments [43]. Despite increased publicity for established preprint review services (such as PREreview [44,45]), there has been a limited use of these platforms [46]. However, independent preprint review projects have arisen whereby reviews are posted in the comments section of preprint servers or hosted on independent websites [47,48]. These more formal projects partly account for the increased commenting on the most high-profile COVID-19 preprints (Fig 4). Although these new review platforms partially combat poor-quality preprints, it is clear that there is a dire need to better understand the general quality and trustworthiness of preprints compared to peer-reviewed articles. Recent studies have suggested that the quality of reporting in preprints differs little from their later peer-reviewed articles [49], and we ourselves are currently undertaking a more detailed analysis. However, the problem of poor-quality science is not unique to preprints and ultimately, a multipronged approach is required to solve some of these issues. For example, scientists must engage more responsibly with journalists and the public, in addition to upholding high standards when sharing research. More significant consequences for academic misconduct and the swift removal of problematic articles will be essential in aiding this. Moreover, the politicisation of public health research has become a polarising issue, and more must be done to combat this; scientific advice should be objective and supported by robust evidence. Media outlets and politicians should not use falsehoods or poor-quality science to further a personal agenda. Thirdly, transparency within the scientific process is essential in improving the understanding of its internal dynamics and providing accountability.

Our data demonstrate the indispensable role that preprints, and preprint servers, are playing during a global pandemic. By communicating science through preprints, we are sharing research at a faster rate and with greater transparency than allowed by the current journal infrastructure. Furthermore, we provide evidence for important future discussions around scientific publishing and the use of preprint servers.

## Methods

### Preprint metadata for bioRxiv and medRxiv

We retrieved basic preprint metadata (DOIs, titles, abstracts, author names, corresponding author name and institution, dates, versions, licenses, categories, and published article links) for bioRxiv and medRxiv preprints via the bioRxiv Application Programming Interface (API; https://api.biorxiv.org). The API accepts a "server" parameter to enable retrieval of records for both bioRxiv and medRxiv. We initially collected metadata for all preprints posted from the time of the server's launch, corresponding to November 2013 for bioRxiv and June 2019 for medRxiv, until the end of our analysis period on October 31, 2020 ($N$ = 114,214). Preprint metadata, and metadata related to their linked published articles, were collected in the first week of December 2020. Note that where multiple preprint versions existed, we included only the earliest version and recorded the total number of following revisions. Preprints were classified as "COVID-19 preprints" or "non-COVID-19 preprints" on the basis of the following terms contained within their titles or abstracts (case insensitive): "coronavirus," "covid-19,"

"sars-cov," "ncov-2019," "2019-ncov," "hcov-19," "sars-2." For comparison of preprint behaviour between the COVID-19 outbreak and previous viral epidemics, namely Western Africa Ebola virus and Zika virus (S1 Fig), the same procedure was applied using the keywords "ebola" or "zebov" and "zika" or "zikv," respectively.

For a subset of preprints posted between September 1, 2019 and April 30, 2020 ($N$ = 25,883), we enhanced the basic preprint metadata with data from a number of other sources, as outlined below. Note that this time period was chosen to encapsulate a 10-month analysis period from January 1 to October 31, 2020, in which we make comparative analysis between COVID-19 and non-COVID-19–related preprints, ($N$ = 44,503), as well as the preceding year from January 1 to December 31, 2019 ($N$ = 30,094), to use as a pre-COVID-19 control group. Of the preprints contained in the 10-month analysis period, 10,232 (23.0%) contained COVID-19–related keywords in their titles or abstracts.

For all preprints contained in the subset, disambiguated author affiliation and country data for corresponding authors were retrieved by querying raw affiliation strings against the Research Organisation Registry (ROR) API (https://github.com/ror-community/ror-api). The API provides a service for matching affiliation strings against institutions contained in the registry, on the basis of multiple matching types (named "phrase," "common terms," "fuzzy," "heuristics," and "acronyms"). The service returns a list of potential matched institutions and their country, as well as the matching type used, a confidence score with values between 0 and 1, and a binary "chosen" indicator relating to the most confidently matched institution. A small number (approximately 500) of raw affiliation strings returned from the bioRxiv API were truncated at 160 characters; for these records, we conducted web scraping using the rvest package for R [50] to retrieve the full affiliation strings of corresponding authors from the bioRxiv public web pages, prior to matching. For the purposes of our study, we aimed for higher precision than recall, and thus only included matched institutions where the API returned a confidence score of 1. A manual check of a sample of returned results also suggested higher precision for results returned using the "phrase" matching type, and thus we only retained results using this matching type. In a final step, we applied manual corrections to the country information for a small subset of records where false positives would be most likely to influence our results by (a) iteratively examining the chronologically first preprint associated with each country following affiliation matching and applying manual rules to correct mismatched institutions until no further errors were detected ($n$ = 8 institutions); and (b) examining the top 50 most common raw affiliation strings and applying manual rules to correct any mismatched or unmatched institutions ($n$ = 2 institutions). In total, we matched 54,289 preprints to a country (72.8%); for COVID-19 preprints alone, 6,692 preprints (65.4%) were matched to a country. Note that a similar, albeit more sophisticated method of matching bioRxiv affiliation information with the ROR API service was recently documented by Abdill and colleagues [51].

Word counts and reference counts for each preprint were also added to the basic preprint metadata via scraping of the bioRxiv public web pages (medRxiv currently does not display full HTML texts, and so calculating word and reference counts was limited to bioRxiv preprints). Web scraping was conducted using the rvest package for R [50]. Word counts refer to words contained only in the main body text, after removing the abstract, figure captions, table captions, acknowledgements, and references. In a small number of cases, word counts could not be retrieved because no full text existed; this occurs as we targeted only the first version of a preprint, but in cases where a second version was uploaded very shortly (i.e., within a few days) after the first version, the full-text article was generated only for the second version. Word and reference counts were retrieved for 61,397 of 61,866 bioRxiv preprints (99.2%); for COVID-19 preprints alone, word and reference counts were retrieved for 2,314 of 2,333

preprints (99.2%). Word counts ranged from 408 to 49,064 words, while reference counts ranged from 1 to 566 references.

Our basic preprint metadata retrieved from the bioRxiv API also contained DOI links to published versions (i.e., a peer-reviewed journal article) of preprints, where available. In total, 22,151 records in our preprint subset (29.7%) contained links to published articles, although of COVID-19 preprints, only 2,164 preprints contained such links (21.1%). It should be noted that COVID-19 articles are heavily weighted towards the most recent months of the dataset and have thus had less time to progress through the journal publication process. Links to published articles are likely an underestimate of the total proportion of articles that have been subsequently published in journals—both as a result of the delay between articles being published in a journal and being detected by bioRxiv and bioRxiv missing some links to published articles when, e.g., titles change significantly between the preprint and published version [25]. Published article metadata (titles, abstracts, publication dates, journal, and publisher name) were retrieved by querying each DOI against the Crossref API (https://api.crossref.org), using the rcrossref package for R [52]. With respect to publication dates, we use the Crossref "created" field which represent the date on which metadata was first deposited and has been suggested as a good proxy of the first online availability of an article [53,54]. When calculating delay from preprint posting to publication dates, erroneous negative values (i.e., preprints posted after published versions) were ignored. We also retrieved data regarding the open access status of each article by querying each DOI against the Unpaywall API (https://unpaywall.org/products/api) via the roadoi package for R [55].

## Usage, altmetrics, and citation data

For investigating the rates at which preprints are used, shared, and cited, we collected detailed usage, altmetrics, and citation data for all bioRxiv and medRxiv preprints posted between January 1, 2019 and October 31, 2020 (i.e., for every preprint where we collected detailed metadata, as described in the previous section). All usage, altmetrics, and citation data were collected in the first week of December 2020.

Usage data (abstract views and PDF downloads) were scraped from the bioRxiv and medRxiv public web pages using the rvest package for R [50]. bioRxiv and medRxiv web pages display abstract views and PDF downloads on a calendar month basis; for subsequent analysis (e.g., Fig 4), these were summed to generate total abstract views and downloads since the time of preprint posting. In total, usage data were recorded for 74,461 preprints (99.8%)—a small number were not recorded, possibly due to server issues during the web scraping process. Note that bioRxiv web pages also display counts of full-text views, although we did not include these data in our final analysis. This was partially to ensure consistency with medRxiv, which currently does not provide display full HTML texts, and partially due to ambiguities in the timeline of full-text publishing—the full text of a preprint is added several days after the preprint is first available, but the exact delay appears to vary from preprint to preprint. We also compared rates of PDF downloads for bioRxiv and medRxiv preprints with other preprint servers (SSRN and Research Square) (S3C Fig)—these data were provided directly by representatives of each of the respective preprint servers.

Counts of multiple altmetric indicators (mentions in tweets, blogs, and news articles) were retrieved via Altmetric (https://www.altmetric.com), a service that monitors and aggregates mentions to scientific articles on various online platforms. Altmetric provide a free API (https://api.altmetric.com) against which we queried each preprint DOI in our analysis set. Importantly, Altmetric only contains records where an article has been mentioned in at least one of the sources tracked; thus, if our query returned an invalid response, we recorded counts

for all indicators as 0. Coverage of each indicator (i.e., the proportion of preprints receiving at least a single mention in a particular source) for preprints were 99.3%, 10.3%, 7.4%, and 0.33 for mentions in tweets, blogs news, and Wikipedia articles, respectively. The high coverage on Twitter is likely driven, at least in part, by automated tweeting of preprints by the official bioRxiv and medRxiv Twitter accounts. For COVID-19 preprints, coverage was found to be 99.99%, 14.3%, 28.7%, and 0.76% for mentions in tweets, blogs, news, and Wikipedia articles, respectively.

To quantitatively capture how high-usage preprints were being received by Twitter users, we retrieved all tweets linking to the top 10 most-tweeted preprints. Tweet IDs were retrieved via the Altmetric API service and then queried against the Twitter API using the rtweet package [56] for R, to retrieve full tweet content.

Citations counts for each preprint were retrieved from the scholarly indexing database Dimensions (https://dimensions.ai). An advantage of using Dimensions in comparison to more traditional citation databases (e.g., Scopus, Web of Science) is that Dimensions also includes preprints from several sources within their database (including from bioRxiv and medRxiv), as well as their respective citation counts. When a preprint was not found, we recorded its citation counts as 0. Of all preprints, 13,298 (29.9%) recorded at least a single citation in Dimensions. For COVID-19 preprints, 5,294 preprints (57.9%) recorded at least a single citation.

## Comments

bioRxiv and medRxiv html pages feature a Disqus (https://disqus.com) comment platform to allow readers to post text comments. Comment counts for each bioRxiv and medRxiv preprint were retrieved via the Disqus API service (https://disqus.com/api/docs/). Where multiple preprint versions existed, comments were aggregated over all versions. Text content of comments for COVID-19 preprints were provided directly by the bioRxiv development team.

## Screening time for bioRxiv and medRxiv

To calculate screening time, we followed the method outlined by Steve Royle [57]. In short, we calculate the screening time as the difference in days between the preprint posting date and the date stamp of submission approval contained within bioRxiv and medRxiv DOIs (only available for preprints posted after December 11, 2019). bioRxiv and medRxiv preprints were filtered to preprints posted between January 1 and October 31, 2020, accounting for the first version of a posted preprint.

## Policy documents

To describe the level of reliance upon preprints in policy documents, a set of policy documents were manually collected from the following institutional sources: the ECDC (including rapid reviews and technical reports), UK POST, and WHO SB ($n$ = 81 COVID-19–related policies, $n$ = 38 non-COVID-19–related policies). COVID-19 policy documents were selected from January 1, 2020 to October 31, 2020. Due to the limited number of non-COVID-19 policy documents from the same time period, these documents were selected dating back to September 2018. Reference lists of each policy document were then text mined and manually verified to calculate the proportion of references that were preprints.

## Journal article data

To compare posting rates of COVID-19 preprints against publication rates of articles published in scientific journals (Fig 1B), we extracted a dataset of COVID-19 journal articles from

Dimensions (https://www.dimensions.ai), via the Dimensions Analytics API service. Journal articles were extracted based on presence of the following terms (case insensitive) in their titles or abstracts: "coronavirus," "covid-19," "sars-cov," "ncov-2019," "2019-ncov," "hcov-19," and "sars-2." Data were extracted in the first week of December 2020 and covered the period January 1, 2020 to October 31, 2020. To ensure consistency of publication dates with our dataset of preprints, journal articles extracted from Dimensions were matched with records in Crossref on the basis of their DOIs (via the Crossref API using the rcrossref package for R [52]), and the Crossref "created" field was used as the publication date. The open access status of each article (S1B Fig) was subsequently determined by querying each DOI against the Unpaywall API via the roadoi package for R [55].

## Statistical analyses

Preprint counts were compared across categories (e.g., COVID-19 or non-COVID-19) using chi-squared tests. Quantitative preprint metrics (e.g., word count and comment count) were compared across categories using Mann–Whitney tests and correlated with other quantitative metrics using Spearman rank tests for univariate comparisons.

For time-variant metrics (e.g., views, downloads, which may be expected to vary with length of preprint availability), we analysed the difference between COVID-19 and non-COVID-19 preprints using generalised linear regression models with calendar days since January 1, 2020 as an additional covariate and negative binomially distributed errors. This allowed estimates of time-adjusted rate ratios comparing COVID-19 and non-COVID-19 preprint metrics. Negative binomial regressions were constructed using the function "glm.nb" in R package MASS [58]. For multivariate categorical comparisons of preprint metrics (e.g., screening time between preprint type and preprint server or publication delay between preprint type and publisher for top 10 publishers), we constructed 2-way factorial ANOVAs, testing for interactions between both category variables in all cases. Pairwise post hoc comparisons of interest were tested using Tukey HSD while correcting for multiple testing, using function "glht" while setting multiple comparisons to "Tukey" in R package multcomp [53].

## Parameters and limitations of this study

We acknowledge a number of limitations in our study. Firstly, to assign a preprint as COVID-19 or not, we used keyword matching to titles/abstracts on the preprint version at the time of our data extraction. This means we may have captured some early preprints, posted before the pandemic that had been subtly revised to include a keyword relating to COVID-19. Our data collection period was a tightly defined window (January to October 2020) which may impact upon the altmetric and usage data we collected as those preprints posted at the end of October would have had less time to accrue these metrics.

## Supporting information

**S1 Fig. Preprints represent a higher proportion of the pandemic-related literature for COVID-19 than previous pandemics, and most articles are open access.** (A) Total number of preprints posted on bioRxiv and medRxiv during multiple epidemics: Western Africa Ebola virus, Zika virus, and COVID-19. The number of preprints posted that were related to the epidemic and the number that were posted but not related to the epidemic in the same time period are shown. Periods of data collection for Western Africa Ebola virus (January 24, 2014 to June 9, 2016) and Zika virus (March 2, 2015 to November 18, 2016) correspond to the periods between the first official medical report and WHO end of Public Health Emergency of International Concern declaration. The period of data collection for COVID-19 refers to the analysis

period used in this study, January 1, 2020 to October 31, 2020. (B) Comparison of COVID-19 journal article accessibility (open versus closed access) according to data provided by Unpaywall (https://unpaywall.org). The data underlying this figure may be found in https://github.com/preprinting-a-pandemic/pandemic_preprints and https://zenodo.org/record/4587214#.YEN22Hmnx9A. COVID-19, Coronavirus Disease 2019; WHO, World Health Organization.
(TIFF)

**S2 Fig. Properties of COVID-19 and non-COVID-19 preprints categorised by preprint server.** (A) Number of new preprints posted to bioRxiv versus medRxiv per month. (B) Preprint screening time in days for bioRxiv versus medRxiv. (C) Number of preprint versions posted to bioRxiv versus medRxiv. (D) License type chosen by authors for bioRxiv versus medRxiv. The data underlying this figure may be found in https://github.com/preprinting-a-pandemic/pandemic_preprints and https://zenodo.org/record/4587214#.YEN22Hmnx9A. COVID-19, Coronavirus Disease 2019.
(TIFF)

**S3 Fig. Additional access statistics for bioRxiv and medRxiv preprints.** (A) Boxplots of abstracts views received by COVID-19 and non-COVID-19 preprints in the same calendar month in which they were posted, binned by preprint posting month. (B) Boxplots of PDF downloads received by COVID-19 and non-COVID-19 preprints in the same calendar month in which they were posted, binned by preprint posting month. (C) Boxplots of total abstract views for non-COVID preprints between January 2019 and October 2020, binned by preprint posting month (D) Boxplots of total PDF downloads for for non-COVID preprints between January 2019 and October 2020, binned by preprint posting month. (E) Comparison of PDF downloads for COVID-19 and non-COVID-19 preprints across multiple preprint servers. Red shaded areas in (C) and (D) represent our analysis time period, concurrent with the COVID-19 pandemic. Boxplot horizontal lines denote lower quartile, median, upper quartile, with whiskers extending to 1.5*IQR. All boxplots additionally show raw data values for individual preprints with added horizontal jitter for visibility. The data underlying this figure may be found in https://github.com/preprinting-a-pandemic/pandemic_preprints and https://zenodo.org/record/4587214#.YEN22Hmnx9A. COVID-19, Coronavirus Disease 2019.
(TIFF)

**S4 Fig. Additional COVID-19 preprint usage data.** (A) Wordcloud of hashtags for the 100 most tweeted COVID-19 preprints. The size of the word reflects the hashtag frequency (larger = more frequent). Only hashtags used in at least 5 original tweets (excluding retweets) were included. Some common terms relating directly to COVID-19 were removed for visualisation ("covid19," "coronavirus," "ncov2019," "covid," "covid2019," "sarscov2," "2019ncov," "hcov19," "19," "novelcoronavirus," "corona," "coronaovirus," "coronarovirus," and "coronarvirus"). (B) Euler diagram showing overlap between the 10 most tweeted COVID-19 preprints, the 10 most covered COVID-19 preprints in the news, the 10 most blogged about preprints, the 10 most commented-upon preprints, and the 10 most cited COVID-19 preprints. The data underlying this figure may be found in https://github.com/preprinting-a-pandemic/pandemic_preprints and https://zenodo.org/record/4587214#.YEN22Hmnx9A. COVID-19, Coronavirus Disease 2019.
(TIFF)

**S1 Table. Descriptive statistics for COVID-19 and non-COVID-19 preprints broken down by server.** COVID-19, Coronavirus Disease 2019.
(XLSX)

**S2 Table. Outputs from mixed-effects regression predicting word count using all bioRxiv preprints.**
(XLSX)

**S3 Table. Outputs from mixed-effects regression predicting word count using only published bioRxiv preprints.**
(XLSX)

**S4 Table. Statistics for first time or previous posting of preprints by senior authors based on country.**
(XLSX)

**S1 Model. Mixed-effects regression models to investigate alternative factors on length of preprints.**
(DOCX)

## Acknowledgments

The authors would like to thank Ted Roeder, John Inglis, and Richard Sever from bioRxiv and medRxiv for providing information relating to comments on Coronavirus Disease 2019 (COVID-19) preprints. We would also like to thank Martyn Rittman (preprints.org), Shirley Decker-Lucke (SSRN), and Michele Avissar-Whiting (Research Square) for kindly providing usage data. Further thanks to Helena Brown and Sarah Bunn for conversations regarding media usage and government policy.

## Author Contributions

**Conceptualization:** Nicholas Fraser, Liam Brierley, Gautam Dey, Jessica K. Polka, Máté Pálfy, Jonathon Alexis Coates.

**Data curation:** Nicholas Fraser, Liam Brierley, Jonathon Alexis Coates.

**Formal analysis:** Nicholas Fraser, Liam Brierley, Jonathon Alexis Coates.

**Investigation:** Nicholas Fraser, Liam Brierley, Gautam Dey, Jessica K. Polka, Máté Pálfy, Jonathon Alexis Coates.

**Methodology:** Nicholas Fraser, Liam Brierley, Jonathon Alexis Coates.

**Project administration:** Jonathon Alexis Coates.

**Resources:** Jessica K. Polka, Jonathon Alexis Coates.

**Software:** Nicholas Fraser, Liam Brierley.

**Supervision:** Jonathon Alexis Coates.

**Validation:** Nicholas Fraser, Liam Brierley, Jonathon Alexis Coates.

**Visualization:** Nicholas Fraser, Liam Brierley.

**Writing – original draft:** Nicholas Fraser, Liam Brierley, Gautam Dey, Jessica K. Polka, Máté Pálfy, Jonathon Alexis Coates.

**Writing – review & editing:** Nicholas Fraser, Liam Brierley, Gautam Dey, Jessica K. Polka, Máté Pálfy, Federico Nanni, Jonathon Alexis Coates.

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
