## [Editor Report · Decision Letter 0]

22 Oct 2020

Dear Dr Coates, 

Thank you for submitting your manuscript entitled "Preprinting the COVID-19 pandemic" for consideration as a Meta-Research Article by PLOS Biology.

Your manuscript has now been evaluated by the PLOS Biology editorial staff, as well as by an academic editor with relevant expertise, and I'm writing to let you know that we would like to send your submission out for external peer review.

Please re-submit your manuscript within two working days, i.e. by Oct 26 2020 11:59PM.

Kind regards,

Roli Roberts

Senior Editor

PLOS Biology

---

## [Decision Letter · Decision Letter 1]

16 Nov 2020

Dear Dr Coates,

Thank you very much for submitting your manuscript "Preprinting the COVID-19 pandemic" for consideration as a Meta-Research Article at PLOS Biology. Your manuscript has been evaluated by the PLOS Biology editors, an Academic Editor with relevant expertise, and by four independent reviewers.

You'll see that all four reviewers are broadly positive about your study, but each of them raises a number of concerns, some of which will need additional analyses (and in some cases data) to address.

IMPORTANT:

a) Three of the four reviewers request that you update your article by considering data for preprints beyond April 30th. Given the potential for further interesting trends preprint deposition and sharing after this date, we think that this is important to address.

b) Given some of the political insights that you discuss, you might find the following recent article about preprint audience segmentation useful. It's by Carlson and Harris, and was published in PLOS Biology just two weeks before you submitted, so you may not have been aware of it: https://journals.plos.org/plosbiology/article?id=10.1371/journal.pbio.3000860

In light of the reviews (below), we will not be able to accept the current version of the manuscript, but we would welcome re-submission of a much-revised version that takes into account the reviewers' comments. We cannot make any decision about publication until we have seen the revised manuscript and your response to the reviewers' comments. Your revised manuscript is also likely to be sent for further evaluation by the reviewers.

We expect to receive your revised manuscript within 3 months. 

**IMPORTANT - SUBMITTING YOUR REVISION**

*Re-submission Checklist*

*Published Peer Review*

*PLOS Data Policy*

*Blot and Gel Data Policy*

Sincerely,

Roli Roberts

Senior Editor,

rroberts@plos.org,

PLOS Biology

REVIEWERS' COMMENTS:

Reviewer #1:

In "Preprinting the COVID-19 pandemic," Fraser et al. present a useful and timely analysis on the fate of preprints during the COVID-19 pandemic. The paper reports that preprints investigating COVID-19 are published at a higher rate than preprints on other topics, with much shorter delay between preprint and publication. The study also shows that COVID-19 preprints are much shorter and highlight interesting differences between papers at bioRxiv and medRxiv, particularly regarding screening time. This work represents an important contribution to our understanding of the role of preprints in the current pandemic and what we might expect from a future crisis. The data collection approach is logical, organized and thorough. However, there are several major issues that I recommend addressing. 

MAJOR POINTS

1. The manuscript's primary dataset is preprints submitted to bioRxiv and medRxiv between 1 Jan 2020 and 30 Apr 2020. The work asks important questions, but, entering the 11th month of a pandemic in which conditions continue to change quickly, the answers presented in the paper may be representative of the pandemic, as trends may have shifted since April. The manuscript would greatly benefit from analyzing preprints from January through at least the summer, when COVID-19 preprint submissions began to decline again: https://github.com/nicholasmfraser/covid19_preprints

2. The results regarding publication rate (and publication delay) would be much more interpretable if the deadline for publication were extended significantly past the deadline for posting the preprint. Currently, the 30 Apr cutoff for preprints appears to be identical to the cutoff for publication—i.e. preprints posted on 27 Apr still count toward the tally of total preprints, but would only count toward the published preprints if they were published within 3 days of appearing online. This has at least two direct effects on the results: First, it artificially deflates the COVID-19 publication rate, particularly relative to the non-COVID-19 publication rate, which is based on a corpus that is growing at a slower (relative) pace. Second, and more importantly, it skews the "time to publication" measurement in favor of preprints that are published quickly: The paper reports that many COVID-19 preprints are published in less than 30 days, but, given the growth pattern of COVID-19 preprints, most papers in their dataset could only have been published in less than 30 days. If the same preprints were evaluated, but the publication cutoff was extended for another 4 months, would the distribution even out? One way to examine this would be to re-evaluate publication-related outcomes after excluding preprints posted close to the publication cutoff. For example, if publication data is analyzed as of 30 Sep, it would be helpful if publication results only considered preprints posted before, say, 1 Aug, even if the other analyses consider preprints posted all the way through September.

3. Several calculations described in the paper appear to be incorrect. While these would almost certainly be caught during re-analysis of an expanded dataset, I wanted to highlight them here as well. I couldn't find code for the analyses, so I made my best guess for how to replicate the tests given the data in the paper's associated GitHub repository.

Line 144: The paper states, "single-author preprints were almost three times more common among COVID-19 than non-COVID-19 preprints." By my count, 205 out of 2527 COVID preprints (8.1%) listed only one author, and 288 out of 12285 non-COVID preprints (2.3%). Dividing the COVID proportion by the non-COVID proportion is 3.46, not "almost three."

Line 191: Using the data in "preprints_full_20190901_20200430.csv," I can't reproduce the reported medians here. When comparing COVID and non-COVID preprints, the text states it's 3432 vs 6143, but I get 3472 vs 6174.

Lines 202-204: The text says that 4% of COVID preprints were published, compared to 3% of non-COVID preprints. However, Figure 2I indicates the COVID publication rate is somewhere around 12 percent, with the non-COVID rate around 6 or 7. The provided data supports the version in the text, so it would be helpful to fix (or explain) the difference here.

Line 210: The paper reports a mean publication delay of 22.5 days for COVID preprints and 48.7 days for non-COVID preprints, but from their data, I get means of 21.0 and 32.7, a much smaller gap. The difference in means is tested using a two-way ANOVA test, but only one interaction term (COVID vs non-COVID) is clear from the description. If "source" (bioRxiv vs. medRxiv) is used as the second term, an ANOVA for "delay_in_days ~ covid_preprint+source" returns non-significant F-values for both source and covid_preprint, suggesting COVID status may not actually affect publication time—a big change from the stated results.

Line 214: The paper describes an ANOVA using publishers, but the degrees of freedom listed (283) suggest they actually used journals, not publishers. It could well just be a typo, but there are 284 journals and only 62 publishers in the dataset. Clarification would be helpful.

Line 270: The text states that 6.7% of non-COVID preprints were mentioned in a news article. Of the 12,285 non-COVID preprints in the analysis period, I only see 83 with at least one mention in a news article, which is 0.68%.

Figure 2: It appears panel 2J has incorrect data in it: It caught my eye because there are papers in the "140 days" bucket, even though the analysis period is shorter than 140 days. Reproducing the panel using the data in the repository shows a different distribution that doesn't go as far to the right.

Figure S2: I'm unable to reproduce panels E and F from this figure using the data provided. As submitted, the panel says Science has published 15 COVID preprints, for example, but I can only find 4. It says the Journal of Infectious Diseases has published 10, but I can only find 4, and so on. This may be caused by the same issue present in Figure 2J. (Incidentally, the bars in panel E are organized alphabetically, which doesn't seem like the order that would be most relevant to readers. Ordering them by value may present the information in a way that's more easily interpretable.)

4. I defer to the editor on whether this is a major issue, but I wanted to highlight several ways in which the current submission doesn't meet my understanding of the PLOS Biology requirements on data availability. First, the referenced GitHub repository provides thorough access to most of the data, but GitHub isn't intended for archival use and isn't included on the journals list of recommended repositories:

https://journals.plos.org/plosbiology/s/recommended-repositories

The paper would benefit from depositing the code somewhere that it could be easily cited and reliably preserved. Personally, I've had a great experience with Zenodo, which has a feature that enables a direct transfer between GitHub and a free Zenodo repo:

https://guides.github.com/activities/citable-code/

Figure S3: I believe the data used for panel C is missing from the dataset. The PLOS guidelines state that "all numerical values used to generate graphs must be provided": https://journals.plos.org/plosbiology/s/submission-guidelines

Similarly, there is no data available to reproduce the results described on lines 233-243. While it looks like the data was not licensed for public release, is it possible anonymized data (e.g. lists of downloads, without any metadata attached) would be allowed, since that's effectively what appears in the figure?

Lines 215-217: The statement that "non-COVID-19 preprints had a 10.6% higher acceptance rate than COVID-19 manuscripts" is poorly supported here. While it's not hard to believe this is an accurate characterization of the data that was provided to the authors, there is no information more specific than "several publishers" about what journals this refers to. In addition, readers are not provided with any data to support the findings, nor, as required in the PLOS guidelines, are they given "All necessary contact information others would need to apply to gain access to the data."

https://journals.plos.org/plosbiology/s/data-availability

MINOR POINTS

Line 86: The paper states that COVID-19 preprints "are reviewed faster than their non-COVID-19 counterparts," but that isn't the only explanation for the observed differences. Given the stakes, it's not impossible that preprint authors were just less likely to post a preprint until they knew they had a done deal at a journal. For authors scared (rightly or not) of getting scooped, posting a preprint 24 hours before your peer-reviewed article goes live may be a way to drum up attention and appear "open" without any risk. Unless there is a way to demonstrate that authors deposit COVID and non-COVID preprints at the same point in the publication process, the statement that they are "reviewed faster" seems to make a large interpretive leap when a phrase like "spend less time on bioRxiv prior to publication" is better justified. This may be a moot point if changes are made regarding Major Issue 2 above.

Line 98: In multiple places (lines 31 and 109, Figure 1B), the text references the number of published papers related to COVID-19, but it's unclear where this information comes from. The legend for figure 1 says "Journal data in (B) is based upon data extracted from Dimensions," but the paper would benefit from elaboration in the Methods section regarding the search strategy and when the search was performed.

Lines 185-187: The paper states that "COVID-19 preprints did not discernibly differ in number of versions compared with non-COVID-19 preprints," using as evidence that both categories have a median of 1. However, Figure 2C shows a noticeable difference in the distributions. Testing the difference between groups using something like the Mann-Whitney test would enable including a definitive statement.

Line 192: The paper states that the difference in preprint length between COVID and non-COVID papers "supports the anecdotal observations that preprints are being used to share more works-in-progress rather than complete stories." However, given the accelerated publication rate of COVID preprints, it seems likely that this could also just indicate that for COVID, the bar for a "complete story" is lower. This isn't a necessary analysis, but if the authors are interested, this section would be improved by an analysis of preprint length among PUBLISHED preprints: Do shorter preprints have a longer delay before publication? If so, I think that would be much more supportive of the idea that people are sharing results as they work. However, if short COVID-19 preprints are published just as quickly as longer ones, that suggests a different story.

Line 195: The text states that the difference in total references between COVID and non-COVID papers reflects "the new, emerging COVID-19 field and dearth of prior literature to reference." However, particularly given the dramatic length difference, maybe shorter, more straightforward COVID preprints simply require less supporting references—if an average non-COVID preprint reports the results of, say, 3 major experiments, it would probably require more background and support than a COVID preprint only reports one experiment. The most straightforward fix for this is to remove some of this over-interpretation, but it may be testable by evaluating something like "references per word."

Line 486: More documentation would be helpful regarding how the preprint publication dates were determined. The text specifies that they were retrieved from Crossref, but the Crossref API provides several different "publication" dates that do not always match. Since there is not an explicit connection between publication date and the date Crossref receives the data, it would also be helpful if the paper specified the dates that the publication data was pulled.

OTHER SUGGESTIONS

The notes below are only intended as friendly suggestions and are not critical to the integrity of the paper.

Lines 27-37: I found it striking that the abstract does not actually describe any results. Readers may be more likely to read on if they're given a better idea of what kind of analysis is to come.

Line 109: The word "published" here is ambiguous: It sounds like the 6,753 preprints included in this total were preprints that were all subsequently published in journals, but, if I'm reading it correctly, this conflates posting to a preprint server with "publishing." Given that the paper deals with preprints that later appeared in journals, it would be beneficial to rephrase this.

Line 155: The phrase "as an expectation" tripped me up here—perhaps "as expected" would be more clear?

337: This sentence suggests the use of preprint servers has been "encouraged by funding bodies requiring COVID-19 research to be open access," but the call to make research open access seems almost separate from the push for people to post preprints—that is, it's not clear to me that posting a preprint prior to publication would satisfy Wellcome's commitment that "all peer-reviewed research publications relevant to the outbreak are made immediately open access." Given that both cited examples strongly encourage preprints but only Wellcome's mentions open-access publication, it may be better to edit this sentence to remove the phrase "requiring COVID-19 research to be open access."

Lines 402-404: Seems like this manuscript is citing its own preprint. This is the first I've seen this, and I'm not sure what is the rationale. If there is an analysis that was included in the preprint version but not the current version, yet still important enough to cite, it might make sense to include the analysis in the current manuscript.

Lines 529-531: The citations of preprints seems very relevant and is important enough to move up to the results section. In addition, the phrase "all preprints" is unclear here, since the manuscript includes references to multiple preprint servers. It appears this refers to the bioRxiv and medRxiv preprints posted in September 2019 or later; it would be helpful to clarify that.

Line 573: It's not clear what method was used for multiple test correction here. The impression that I get from the multcomp documentation is that the glht.summary() function has multiple sophisticated options. It's possible I'm misreading this, but more clarity would be appreciated.

Figure 1: The y-axis label in panel B looks at first glance like it may refer to a ratio, though I think it actually means "articles OR preprints." Would "Manuscripts" be more clear as a label?

Figure 2: Panel B has been effectively scrambled by the lopsided number of preprints processed between medRxiv and bioRxiv, data that is much better visualized in Figure S2 (panel B). Right now, Figure 2B makes it look as if COVID preprints took far longer than usual to be screened, while the results (and Figure S2b) show that there wasn't a big COVID effect, and it's just that COVID preprints were more likely to show up on medRxiv, where screening always takes longer. I suggest replacing Figure 2B with Figure S2B, which is a little more cluttered but far more interpretable.

Figure 2D contains so much interesting data, but is almost impossible to read because of the delineation between first-time authors and returning ones. This panel may be better as a supplementary figure. At the author's discretion, I'd suggest making a scatter plot similar to the one in Figure S2C, plotting COVID vs. non-COVID percentages, leaving the distinction of first-time authors to a supplement.

Panel 2I seems to be an unnecessary use of space—I think readers can conceptually compare one number against another, slightly smaller number without a picture.

Reviewer #2:

[identifies himself as Euan Adie]

Fraser et al. have produced a thorough, detailed analysis of preprints relating to COVID-19 in 2020 and compared them to non-COVID preprints in the same period and earlier.

I'm impressed by the range of the data examined and by the quality of the associated, documented code for the analysis the authors have placed on github, though I didn't re-run the analysis for myself.

We've unquestionably seen a change in the use of preprints during the pandemic and the breadth & wide scope of this study makes it novel and significant enough for publication.

That said I did find myself wanting a clearer picture of what some of the data means, particularly around whether or not how many COVID-19 preprints are genuine work of a standard equivalent to what would normally be submitted to preprint servers vs more lightweight but still high quality articles vs opportunistic "spam" articles. The manuscript touches on some of these points but left me unclear.

My comments in order of appearance rather than importance, (6) is the only revision I'd consider essential:

1) COMMENT Line 70: preprints aren't only -scientific- manuscripts, you mention humanities etc. preprints later on

2) COMMENT Line 80: first time preprints have been widely used to communicate during epidemic: "widely" is doing a lot of work here… as we don't know e.g. what proportion of all researchers working on Zika used preprint servers. Maybe better to say widely used outside of specific communities?

3) COMMENT Line 109 - more than 16,000 articles published, a large proportion preprints: as a number of preprints go on to become published articles it'd be good here to highlight the number that are preprints OR published versions of preprints rather than just the former.

4) QUESTION Line 112 -SSRN - I'm not familiar with the SSRN workflow but do notice that The Lancet "First Look" papers live there https://www.ssrn.com/index.cfm/en/the-lancet/ … were these treated as preprints, and did any other medical journals introduce "one click" ways to deposit papers as preprints in 2020? Basically were any new workflows introduced that might have influenced author choices?

5) COMMENT Line 181 - different types of license being adopted for COVID-19 work: I'd be interested in some brief discussion around why this might be. e.g. are there pharma collaborations or is it connected to how new to preprints authors are? It doesn't gel 100% with authors wanting to be as open as possible - maybe it's just speed that's important to them?

6) ESSENTIAL SUGGESTED REVISION Line 193: support anecdotal evidence that preprints are being used to share works in progress: I'd really like to see this expanded upon here or in the discussion, as it relates also to the quality question you raise around line 201 and later on too… it seems like it would make interpreting other aspects of the data easier. Specifically, does the data suggest that (a) COVID-19 preprints are mostly opportunistic short works or genuinely works in progress, with only one version, that then get submitted to journals with few changes (which may explain the lower acceptance rate, and is what's implied at first)? or (b) do they usually undergo significant changes between preprint and final published version (on line 401 it's asserted that the preprints are of relatively good quality, because the acceptance rate is only a little lower)? In the latter case authors may not engaging with versioning or perhaps publishers have lowered standards for COVID related work, which would be good to know. You start addressing the opportunistic part by looking at changing fields which I think is a good start. I realize that as you say it's very difficult to assess the "quality" of a preprint… perhaps you could get a feel for things by assessing the number of changes in the final published version, for a random subset of the articles?

7) SUGGESTED REVISION: Line 194: COVID-19 preprints have fewer references: are you controlling for number of words? If the articles are 44% shorter it stands to reason that they should also have fewer references, it may be that the articles are more focused or just don't have scope for many citations. I don't think we can say that it reflects only the dearth of prior literature.

8) SUGGESTED REVISION: Line 208: we see faster publication times for COVID-19 preprints: again, would be interesting to see this controlled by article length. Shorter articles with clearer, short hypotheses and opinion pieces will be easier to review and to a certain extent copyedit than longer, data heavy papers. 

9) QUESTION: Line 219: was there any anecdotal evidence from MedRxiv / bioRxiv about where the abstract views were coming from? For example direct links from the websites of public health bodies.

10) COMMENT: Line 339: I think it's too early to say if the change is permanent or related to the specific circumstances of the pandemic

11) SUGGESTED REVISION: Line 376: Marked change in journalistic practice: I suspect this is correct, but it's hard to say without data on *why* papers were picked up more by the new: it could also be because university press offices that were previously only worked with press officers at high impact journals have suddenly become interested in COVID-19 preprints and so scan bioRxiv / medRxiv (or perhaps medRxiv has started reaching out to journalists directly?) and that some researchers are very keen to see public engagement around their COVID work. There are few science journalists and it is rare for journalists 

Reviewer #3:

The manuscript by Fraser et al. is very interesting inasmuch as it documents how many manuscripts on COVID-19 have been shared during the early months of the pandemic. However, the manuscript by Fraser et al. could be easily misread at COVID-19 being special and preprints fulfilling a unique need of the scientific community. While the authors are transparent in communicating their potential conflicts of interest, the latter framing would not seem necessary for the manuscript to be interesting.

My main concerns are well encapsulated in the statement of the abstract "Although the last pandemic occurred only a decade ago, the way science operates and responds to current events has experienced a paradigm shift in the interim. The scientific community responded rapidly to the COVID-19 pandemic, releasing over 16,000 COVID-19 scientific articles within 4 months of the first confirmed case, of which 6,753 were hosted by preprint servers.". This statement and similar impactions throughout the manuscript suggest that the way science operated changed and that this was tied to preprints.

However, I do not see the manuscript of Fraser et al. as providing evidence for a change in the way science is conducted aside from acknowledging that the volume of research dedicated to COVID-19 is large, and larger than for past pandemics. On a philosophical note, the manuscript does not follow's Kuhn's definition of paradigm shift (that would be more similar to a declination). On a scientometric level, biomedical scientists have already been very responsive toward epidemics and given SARS, MERS and other global threats a disproportional share of their attention. This could be seen, for instance, by determining the citation metrics of publications (controlled for years) for publications in MEDLINE of each individual MeSH term. Such an analysis places MeSH terms corresponding to emergent pathogens as the most-cited MeSH terms of at least the last ~15 years, generally even at the peak position of all MeSH terms (if one excluded MeSH terms occurring only in a handful of manuscripts). One may interpret this as no single set of topics of biology having received as much interest of scientists as pathogens causing pandemics. An alternative reading of the manuscript of Fraser et al. could thus be that scientists did - and do - redirect their attention toward emergent pathogens and pandemics - but that the volume of research on COVID-19 is higher than for other emergent pathogenes/epidemics due to other reasons (e.g.: fraction of people in research-heavy countries that are affected?). Likely, one may conclude that preprints have not been necessary for emerging pathogens to draw more attention among scientists than any other topic.

Other comments:

When comparing COVID-19 preprints against others it remains unclear from the methods section and text, whether all comparisons (e.g.: also license type, number of versions of preprints, lengths of texts) are restricted to preprints posted in the same observational period as COVID-19 preprints (which appears introduced as a statement in the context of Fig 2I). In case that the range of dates allowed for non-COVID-19 preprints differed, the interpretation surrounding many elements of Figure 2 might change.

The authors show differences in reception of COVID-19 vs non-COVID-19 manuscripts (Figure 3), which they interpret as "extensive access". While they rule out some possible alternative scenarios, it remains unclear to which extent this reception is driven by scientists using preprints differently in the context of COVID-19 vs. preprint servers (or secondary sites, such as covidpreprints run by one of the authors) prioritizing the visibility of COVID-19 manuscripts over others (e.g.: bioRxiv has a bold red link to "COVID-19 SARS-CoV-2 preprints, suggesting that there are active efforts to help COVID-19 preprints to gain more visibility). Maybe there is data to make a more stringent statement, otherwise, I would recommend rephrasing.

The data as shown in panels Fig 4A-E provides very little information not contained in the text since distributions are similar, sometimes have the same median, and there are many dots - but it remains unintuitive for reader if there would be a relative change (as total number of dots could be dominated by total number of preprints in each category). Possibly more information could be conveyed by making a cumulative plot (or survival analysis) with increasing values (now y) as thresholds (on x), and plotting two different lines (COVID-19 and non-COVID 19). As an additional, related challenge around these panel, it appears that likely the statistical tests given in the main text should be replaced by non-parametric test, or replaced by tests that do not test differences in the centrality (as dynamic range is small) - but instead test (e.g.: via Fisher's exact test) whether proportions of preprints with at least one y-value (e.g.: one citation) would differ between COVID-19 and non-COVID-19 preprints.

For Figure 4G,H it is unclear whether the inferred statement on higher correlations among COVID-19 truly reflect different correlations, or a higher number of preprints with non-zero values among the COVID-19 group.

The discussion spares possible criticisms around preprints. First, the tournament-type economics of science that requires scientists to accumulate reputation may force researchers to publish on preprint servers due to the risk of being scooped rather than because of their perception of the usefulness of preprint servers, and thus contribute to a research culture that could be perceived unfavorably (e.g.: along "publish-or-perish"). Second, scientific disciplines can slow down in their rate of innovation if they grow too big (Chu et Evans, 2018, ScArXiv). In this sense a larger volume of publications (in manuscripts, but further increased by preprints) may be expected to lead to more conservative research and rather limit the overall progress of scientific fields (besides size).

The discussion section misses the possibility that preprint servers are not neutral services, but themselves act in a way that could prioritize COVID-19 (e.g.: by highlighting COVID-19 publications as bioRxiv does).

Minor:

Extending the analysis beyond April 30th would be interesting as journals and the scientific community had more time to adopt. Based on Figure 3A, B, which shows diminishing differences between COVID-19 and non-COVID-19 preprints over time, it remains unclear whether the findings reported by the authors throughout the manuscript only refer to the first weeks of a pandemic (a time-period that would be very important, and where preprints might be particularly relevant), or a "shift" as they imply in the discussion section.

"Escalating demands made by reviewers and editors are lengthening the publication process still further [8,9]." isn't backed up by the references, and might rather mirror a common perception, and topic of further study. 

While it would seem unlikely to change the overall findings, the comments on other RNA viruses might be incomplete as the keyword-based matching used by the authors uses fewer synonym than for COVID-19 and excludes most synonyms provided by NCBI Taxonomy, which is a reference database for the nomenclature of organisms.

The analysis of the paragraph between lines 135-141 visually appears at odds with the referenced panels, Figure 2B, and S2B, where non-COVID-19 preprints appear to have been screened more rapidly than non-COVID-19 preprints. Particularly, there appear to be more non-COVID-19 preprints with a screening time of 0 or 1 days. As the distributions of screening times are not normally distributed (and cannot be close to 0 as there is no negative screening time), providing the median rather than mean - and doing so separately for bioRxiv and medRxiv - and providing a fitting non-parametric test (e.g.: ranksum) could more accurately describe the overall trends in the data.

The analysis of the subsequent paragraph, between lines 142-146, the additional variability in team size, a half sentence which separates bioRxiv vs. medRxiv may help to clarify how much of the variability would stem from different team sizes in different academic fields (e.g.:, molecular biology vs. clinical research). 

"Additionally, India had a higher representation among COVID-19 authors specifically using preprints for the first time compared to non-COVID-19 posting patterns." The visual impression from the figure is that the absolute numbers for India are very small, opening the possibility that the difference is not statistically significant. Adding a statistic test to the statement would prevent those thoughts.

Within Figure S2, panel D appears on the left-hand side from panel C, which is opposite to the reading direction (left to right) common in most scientific publications.

The discussion section claims a cultural shift regarding media. However, it remains unclear whether there was a long-lasting effect, as implied through "shift", which remained after April 2020. Further it remains unclear whether there was a conceptual change in the way media would operate that has been enabled by preprints, or whether also in the early days of other epidemics media used essentially any source outside of scientific journals to obtain information that they would include in their coverage (e.g.: interviews with scientists, reports of local health agencies….).

The statement on politicization on science needing to be "prevented at all costs" could be a little bit more specific to avoid reading it in a possibly unintended manner. For instance, one could argue that arguments among politicians should be based on science, that scientists should focus their research on problems identified by societies and their representatives, and that academics should also care about gender or racial injustice that is present in their societies to avoid that these are manifested through educational systems.

Very, very minor point of curiosity - please fully ignore for any practical concerns unless already considered somehow by authors: The manuscript very understandably focuses on academic science. Would the findings also extend to non-academic preprint servers that scientists should ignore for many good reasons, such as viXra.org?

Reviewer #4:

This paper focuses on summarizing various attributes, bibliometrics, and altmetrics of preprints pertaining to the COVID-19 pandemic. Overall, I think the manuscript is thoughtful and thorough, and provides a timely overview of how the pandemic has impacted scientific publishing and vice versa. Even beyond the pandemic, this should prove to be a useful point of reference in the ongoing debate surrounding preprints, open science, and peer review.

While the descriptive statistics and univariate tests provide a nice backdrop for thinking about the unprecedented changes to publishing practices induced by COVID-19, I'd like to see the authors attempt to address some more challenging hypotheses and apply some slightly more sophisticated multivariate statistical analyses to support their conclusions.

MAJOR COMMENTS

The authors allude to "anecdotal observations that preprints are being used to share more works-in-progress" (line 192) as the reason COVID-19 preprints tend to be shorter in length--is this based on the authors' own anecdotal evidence, or are there references that can be cited here? "Works-in-progress" implies the research published was not as rigorous as in non-COVID-19 research--although there are certainly examples where results in a preprint were "half-baked", this scenario should be differentiated from studies in which authors are simply sharing results incrementally in short reports rather than waiting to accumulate multiple results before sharing. This practice is something various publishers have tried to promote over the years (e.g., Cell Reports https://www.cell.com/cell-reports/aims)--perhaps the authors could tease out the "work in progress" versus "short report" hypotheses by testing if the word count of COVID-19 preprints is associated with higher rates of publication or faster turnaround in peer-reviewed journals.

I can think of a few other explanations for the shorter length of COVID-19 preprints that could be tested, e.g., perhaps epidemiology papers tend to be shorter than papers in other fields of study and epidemiological studies are overrepresented among the COVID-19 preprints. Similarly, the authors mention elsewhere that relatively more COVID-19 preprints tend to have only a single author, which could also partially explain the shorter length, since there are ostensibly fewer person-hours invested in the writing than a multi-author study. There might also be cultural differences that contribute to paper length--do authors from China or India (the two countries noted as having the greatest increase in representation among COVID-19 preprints) tend to write shorter papers overall? Later, the authors recognize that COVID-19 preprints contain fewer references, which could itself contribute to the shorter length, as there is less need to situate new results against existing literature (in which case we might expect these preprints to have gotten longer as the pandemic has progressed). It should be straightforward to apply a regression model to assess the relationship between paper length and COVID-19 focus, adjusting for topic, author count, author country, date posted, etc.

In the section "Extensive access of preprint servers for COVID-19 research", how much are the average number of abstract views/downloads influenced by outliers like the Santa Clara seroprevalence study and the withdrawn "uncanny inserts" study? More generally, are abstract views/downloads strongly correlated with attention on social media?

Lines 267-270: it would be good to provide some numbers here on the total number of original tweets and hashtags analyzed. Also, take some space to elaborate on why hydroxychloroquine is a controversial topic and why certain conspiracy theories have latched onto these top 10 most tweeted preprints. The wordcloud in Supplemental Fig 4A shows some extraordinary evidence of politicization that isn't mentioned, including the QAnon conspiracy theory ("qanon" and "wwg1wga"), xenophobia ("chinazi"), and US-specific right-wing populism ("maga", "foxnews", "firefauci") (I also think this figure could be moved to the main text). Given that the authors don't shy away from denouncing the politicisation of science as "a polarising issue [that] must be prevented at all costs" (line 409), this section feels much too short in its current form. 

The paper makes several references to research that is "poor-quality" or "controversial" but does not rigorously define or classify such preprints. With all of the data at hand, a cool deliverable might be to isolate particular attributes associated with low quality or propensity for controversy. Even some simple descriptives of a curated subset of such preprints would be interesting.

I understand the following request might not be feasible, so consider it optional, but it would be great to see the results of this paper updated to include preprints published more recently than April 30--there's a full 6 months of data that are ignored (spanning the first big peak of cases in the US in July and the ongoing second wave), and there are potentially some really interesting stories to tell--not just about the overall characteristics of COVID-19 preprints, but how they have evolved over time.

MINOR COMMENTS

Fig 1a: since case and death counts are shown together on the same panel, this figure would be more readable with the y-axis on a log scale

Fig 1b: Since all of the other figures use the same color scheme for COVID-19 preprints vs non-COVID preprints, it would be better to use a different color scheme to describe preprints vs. journal articles here.

Line 55: closing parenthesis should come after "case"

Line 70: "certified" has strong connotations--better to just say preprints have not gone through formal peer review yet

Line 98: Maybe say "*at least* 186 articles," unless the authors are certain this is an exhaustive count

Line 109: Were any preprints that went on to be published in peer-reviewed journals double-counted among the 16,000 COVID-19 articles mentioned here?

Line 121: spell out OASPA acronym

---

## [Decision Letter · Decision Letter 2]

1 Mar 2021

Dear Dr Coates,

Thank you for submitting your revised Research Article entitled "Preprinting the COVID-19 pandemic" for publication in PLOS Biology. I have now obtained advice from three of the original reviewers and have discussed their comments with the Academic Editor. 

Based on the reviews, we will probably accept this manuscript for publication, provided you satisfactorily address the remaining points raised by the reviewers. Please also make sure to address the following data and other policy-related requests.

IMPORTANT:

a) Please attend to the remaining requests from reviewers #1 and #3.

b) Please could you choose a more informative Title. We suggest something like "Analysing the role of preprints in the dissemination of COVID-19 research and their impact on the science communication landscape," but please see the relevant comment by reviewer #1 and feel free to choose something that you think reflects the analysis and findings.

c) Please supply a blurb in the box in the submission form.

d) Many thanks for providing the data and code so fully in Github and Zenodo. Please could you cite the URLs/DOIs clearly in all relevant main and supplementary Figure legends (e.g. "The data underlying this Figure may be found in https://github.com/preprinting-a-pandemic/pandemic_preprints and https://zenodo.org/record/4501924").

We expect to receive your revised manuscript within two weeks. 

*Published Peer Review History*

*Early Version*

Sincerely,

Roli Roberts

Senior Editor,

rroberts@plos.org,

PLOS Biology

DATA NOT SHOWN?

- Please note that per journal policy, we do not allow the mention of "data not sown", "personal communication", "manuscript in preparation" or other references to data that is not publicly available or contained within this manuscript. Please either remove mention of these data or add figures presenting the results and the data underlying the figure(s).

REVIEWERS' COMMENTS:

Reviewer #1:

This revision by Fraser et al. is an interesting analysis and a valuable contribution to the field. They have addressed all of my major concerns, and the expansion of the dataset through October has enabled them to provide a much more comprehensive characterization of the relevant patterns. There are a few minor issues left:

MINOR NOTES:

I defer to the editor on whether this is a concern, but the current title, "Preprinting the pandemic," does not seem as descriptive as it could be. It reads as a feature headline but doesn't reveal any of the findings, nor does it describe the analysis in a useful way.

Line 218: The paper states, "Critics have previously raised concerns that by forgoing the traditional peer-review process, preprint servers could be flooded by poor-quality research." A citation would be helpful here.

Lines 245-253: A lot of space and effort is spent here explaining that downloads for individual preprints taper off over time—this would be a valuable visualization to add, perhaps as a panel in Figure 5. It would help evaluate whether the average "debut month" was shrinking over time, which could indicate popular interest in COVID is waning. An example of a "downloads in first month" figure is available as Figure 2, figure supplement 3(a) in Abdill & Blekhman 2019 [1]—the x-axis could be something like "Month posted," and the y-axis would be "Downloads in first month." Using a visualization such as a box-and-whisker plot could illustrate how many downloads were received in, say, March, for preprints posted in March, followed by downloads in April for preprints posted in April, and so on.

Figure 1: In panel C, it's difficult to see differences between the smaller servers, particularly because the number of segments means the colors are all very similar. It might be better to push the 7 smallest ones into the "other" bucket. This is not a critical issue and we leave it to the authors and editors whether to make this change.

Figure 3: In panel B, it's difficult to compare the two categories because of the dramatically different counts. The distributions (as in panel A) would be much more informative, particularly because the difference in distributions seems to be the most relevant result.

Figure 6: The content of Panel E is limited by the poor match between the data and the scale of the y-axis. While it's logical that all panels would have the same y-axis, the primary comparison for readers doesn't seem to be between panels, but within them. The authors might consider altering the y-axis of this panel to make the differences easier to see.

Several panels in figures 2, 4, 5 and 6 appear to use German abbreviations for month names, while others use English abbreviations. I'm not aware of specific language requirements from the journal, but consistency would be helpful.

References:

[1] Abdill & Blekhman 2019. https://doi.org/10.7554/eLife.45133

Reviewer #3:

Fraser et al. present a series of good improvements for an already interesting manuscript.

I hope that prior publication, they could remove the last remnant of my prior main criticism - namely the reference to a "paradigm shift" contained in the abstract. The phrase "cultural shift" which they now use in the discussion section is very appropriate and fitting as their findings document an (important) shift in publication practices. In contrast, the original meaning of "paradigm shift" within studies of science refers to the way scientists probe phenomena. Kuhn emphasized this point in the foreword of later editions of The Structure of Scientific Revolutions as he noted that his phrase of "paradigm shift" had already become read in a more general and unintended manner by some. Similarly - at least for genes - research on COVID-19 appears to often use research patterns of the past (Stoeger et Amaral, eLife 2020). This would argue against a "paradigm shift" in its original sense.

Other:

Figure labels for months use German abbreviations.

While the authors have now greatly clarified points about the disproportional usage of preprint servers for COVID-19 this argument could possibly be extended in a half sentence that extends beyond their share in preprint servers, relative to other pandemics, toward the comparison of the share of COVID-19 in preprint servers against their share in journals. Such an option would seem supported by an approximate estimate of there being 100,000 COVID-19 publications according to LitCovid, and there being around 2.5 million papers added to MEDLINE every year (thus the ~25% of COVID-19 publications in preprint servers would exceed the ~4% anticipated for journals).

The reformulated text around lines 251 now seems to yield the originally intended message of the authors, which I had not noted in the initial review. As the analysis of this paragraph now stands, I would see their current suggestions as only one of two different options. The other option would be that COVID-19, but not non-COVID-19, preprints are subjected to an additional time-dependent factor (which one could suspect to be the changes of the number of publications available in the literature through journals). I believe that this point could be clarified by formulating the sentence again in a more general manner, or by doing an additional analysis (e.g. as the authors will have at least two time-stamped data queries, with one corresponding to the original submission, and the other corresponding to the update of the revision).

Reviewer #4:

The authors have thoroughly addressed all of my previous comments.

---

## [Editor Report · Decision Letter 3]

8 Mar 2021

Dear Dr Coates,

On behalf of my colleagues and the Academic Editor, Ulrich Dirnagl, I'm pleased to say that we can in principle offer to publish your Research Article "The evolving role of preprints in the dissemination of COVID-19 research and their impact on the science communication landscape" in PLOS Biology, provided you address any remaining formatting and reporting issues. These will be detailed in an email that will follow this letter and that you will usually receive within 2-3 business days, during which time no action is required from you. Please note that we will not be able to formally accept your manuscript and schedule it for publication until you have made the required changes.

PRESS: We frequently collaborate with press offices. If your institution or institutions have a press office, please notify them about your upcoming paper at this point, to enable them to help maximise its impact. If the press office is planning to promote your findings, we would be grateful if they could coordinate with biologypress@plos.org. If you have not yet opted out of the early version process, we ask that you notify us immediately of any press plans so that we may do so on your behalf.

Thank you again for supporting Open Access publishing. We look forward to publishing your paper in PLOS Biology. 

Sincerely, 

Roli Roberts

Roland G Roberts, PhD 

Senior Editor 

PLOS Biology